# SEPAL: Scalable Feature Learning on Huge Knowledge Graphs

## Abstract

Knowledge graphs accumulate information about more and more entities of the world. Much research is conducted to improve embedding models that capture this information and give useful node features in many downstream applications. However, most current methods are hard to scale to large knowledge graphs, partly because GPU memory is too small to hold the embeddings of many entities –YAGO4 has 67M entities. To scale existing embedding models on modest hardware, we introduce SEPAL: Scalable Embedding Propagation Algorithm for Large knowledge graphs. The key idea of SEPAL to reduce compute is to only optimize embeddings on a core subset of entities, those that come with much more information than others. Then SEPAL propagates these embeddings to the rest of the graph with message passing, but no explicit optimization. To enable efficient message passing, we break down large graphs into well-connected subgraphs that fit in GPU memory using a new algorithm called BLOCS: Balanced Local Overlapping Connected Subgraphs. We evaluate SEPAL on five different knowledge graphs for four downstream regression tasks. We show that SEPAL outperforms alternative on downstream tasks, while providing a $43\times$ speedup to its base embedding algorithm. Moreover, outside the core subgraph, embeddings obtained by message passing are not degraded compared to traditional methods, demonstrating the validity of SEPAL's propagation.

## 1 Introduction

Relational data gathers various information on different objects across multiple tables. Modern general-purpose knowledge graph push the agenda to describe an increasingly large fraction of the entities of the world: Wikidata (Vrandečić & Krötzsch, 2014) describes as of 2024 109M entities, and YAGO4 gives a curated view on 67M entities (Pellissier Tanon et al., 2020). For machine learning and artificial intelligence, capturing general knowledge opens an old promise of making tasks easier via this knowledge (Lenat & Feigenbaum, 2000). Integrating this information into machine learning does raise the challenge of assembling features from multiple tables. For this purpose, graph-embedding methods provide node features that can improve downstream learning task (Grover & Leskovec, 2016; Cvetkov-Iliev et al., 2023; Robinson et al., 2024). Increasingly sophisticated embeddings models (Bordes et al., 2013; Yang et al., 2014; Balazevic et al., 2019, ...) help produce embeddings that better capture the relational aspect of the data, which is important for downstream tasks (Cvetkov-Iliev et al., 2023).

The size of large knowledge graphs is exploding: Wikidata gains 20M entities yearly (Wikimedia). This enables the exciting prospect of general feature enrichment: given a downstream table, entities can be automatically linked to the knowledge graph (Mendes et al., 2011; Foppiano & Romary, 2020; Delpeuch, 2019), and node features could be inserted in the table to facilitate machine learning task. The more entities in the knowledge graph, the more this process provides value to the downstream analysis. And yet, there can be a disconnect between the growth of knowledge graphs, and the fact that the increasingly complex embedding models tend to be less tractable, and are typically demonstrated on comparatively small graphs, often subsets of real-world graphs such as FB15k (15k entities from Freebase) or WN18 (40k entities from WordNet), 3 orders of magnitude smaller than modern general knowledge graphs, or industrial knowledge graphs (Sullivan, 2020). One roadblock to scaling knowledge-graph embeddings is that, with many entities, the embeddings no longer fit in the memory of GPUs. The typical answer to this challenge is distributed computation across GPUs,

explored by PyTorch-BigGraph (Lerer et al., 2019) and many more (Zheng et al., 2020; Mohoney et al., 2021; Zhu et al., 2019; Dong et al., 2022; Ren et al., 2022, ...). This comes with sizeable engineering and computational cost, as the graph is fit piece by piece. The multi-GPU requirement is a challenge for non-profits such as the Wikimedia foundation, in charge of Wikidata, given that the embeddings should be recomputed regularly to incorporate newly-added entities.

Here we show how to scale most knowledge-graph embedding methods with little computational resources. Our goal is to bridge the gap between advanced embedding approaches and huge general-purpose knowledge graphs that strive to gather all human knowledge at once. For this, we leverage fundamental structure in knowledge graphs: a small set of "core" entities come with much more information than the others. Our method, SEPAL (Scalable Embedding Propagation Algorithm for Large knowledge graphs), is 'pluggable' to any embedding model that, at triple-level, models the tail embedding as a relation-specific transformation of the head embedding. There are two technical contributions that enable SEPAL to bring scalability. *1)* We show that good embeddings on a small core subset of entities can be propagated to give good embeddings for the full graph. The challenge here is to maintain the relational geometry. *2)* We devise an algorithm called BLOCS to break down a huge knowledge graph into overlapping subsets that fit in GPU memory. Here, the challenge lies in the scale-free and connectivity properties of a large knowledge graph: some nodes are connected to a significant fraction of the graphs, while others are very hard to reach.

We start by reviewing related work. Then, section 3 describes our contributions. In section 4 we evaluate SEPAL's performance on knowledge graphs of increasing size between YAGO3 (Mahdisoltani et al., 2014) and YAGO4 (67M entities, Pellissier Tanon et al., 2020); we study the use of the embeddings for feature-enrichment on four downstream machine learning tasks, showing that SEPAL makes embedding methods much more tractable while generating better embeddings for downstream tasks. Empirical findings show that:

1. propagating embeddings to outer entities with message passing does not lead to a performance loss and gives orders of magnitude speed-ups compared to full optimization;
2. this approach can be scaled to very large knowledge graphs on modest hardware.

## 2 RELATED WORK: EMBEDDING AND SCALABILITY IN KNOWLEDGE GRAPHS

Knowledge graphs are multi-relational graphs describing knowledge in an entity-relation model. Knowledge graphs store information as triples $(h, r, t)$, where $h$ is the head entity, $r$ is the relation, and $t$ is the tail entity.

### 2.1 GRAPH-EMBEDDING METHODS

Graph embedding methods learn low-dimensional (typically $d = 100$ or $200$) vector representations for the entities and relations. Knowledge-graph embeddings are directly related to the more general graph embedding literate, learning representations for graph nodes but not for the relations. Different embedding methods naturally lead to different structures of the embedding space.

**Global methods** Global methods can be formulated at the graph level, typically using the adjacency matrix $A$. A first family of methods performs explicit **matrix factorizations** on matrices derived from the adjacency matrix, for instance GraREP (Cao et al., 2015) or NetMF (Qiu et al., 2018). These methods output close embedding vectors for nodes with similar neighborhoods.

As the adjacency matrix does not represent the edge-type information of multi-relational graphs, it may be preferable to use a $\{0, 1\}$-valued third-order tensor, and correspondingly **tensor factorization** approaches such as canonical polyadic decomposition (Hitchcock, 1927), Tucker decomposition (Tucker, 1966), or more recently RESCAL (Nickel et al., 2011). Unlike other global methods, these methods compute embeddings for both entities and relations. However, they limit the relational model to multiplicative interactions between entities and relations embeddings.

To avoid relying on –potentially costly– optimization, another strategy is to compute **random projections**. Indeed these give very cost-effective approximations of the pairwise distances (Dasgupta & Gupta, 2003). FastRP (Chen et al., 2019) proposes a scalable approach, with a few well-chosen applications of the adjacency matrix on a random projection matrix.

**Local methods** Local methods formulate an optimization on triples rather than on the matrices or tensors. They define a scoring function $f(h, r, t)$ to represent the plausibility of a triple given the embeddings $\theta_h, \theta_r, \theta_t$ of the entities and relation. The embeddings are optimized by stochastic gradient descent to maximize the score of positive triples, and minimize that of negative ones.

**Skip-gram negative sampling** (SGNS), behind word2vec (Mikolov et al., 2013), has been shown to perform an implicit factorization (Levy & Goldberg, 2014). It has been adapted to graphs: Deep-Walk (Perozzi et al., 2014) and node2vec (Grover & Leskovec, 2016) perform random walks on the graph to generate "sentences" fed to word2vec. Here the scoring function between two nodes $h$ and $t$ is simply $f(h, t) = \theta_h \cdot \theta_t$. RDF2vec adapts this framework to multi-relational graphs by adding the relations to the generated sentences (Ristoski & Paulheim, 2016).

**Triple-based methods** design scoring functions to model the relations as geometric transformations in the embedding space. A seminal model is TransE (Bordes et al., 2013), modeling relations as translations in the embedding space. Many of these models can be framed as:

Scoring function
$$f(h, r, t) = -sim(\phi(\theta_h, \theta_r), \theta_t) \tag{1}$$

where $\phi$ is a model-specific relational operator, and $sim$ a similarity function. These models strive to align, for positive triples, the tail embedding $\theta_t$ with the "relationally" transformed head embedding $\phi(\theta_h, \theta_r)$. The challenge is to design a clever $\phi$ operator to model complex patterns in the data –hierarchies, compositions, symmetries... Indeed some relations are one-to-one (people only have one biological mother), well represented by a translation, while others are many-to-one (for instance many person were `BornIn` Paris), calling for $\phi$ to be a contractive operation (Wang et al., 2017). Many models explore different parametrizations, among which MuRE (Balazevic et al., 2019), RotatE (Sun et al., 2019) or QuatE (Zhang et al., 2019) have good performance (Ali et al., 2021b). This framework also includes models like DistMult (Yang et al., 2014), ComplEX (Trouillon et al., 2016) or TuckER (Balažević et al., 2019), that implicitly perform tensor factorizations.

Table 1: Expression of $\phi$ in some embedding models. $\odot$ denotes the Hadamard product, $\otimes$ the Hamilton product, and $\times_i$ the tensor product along mode $i$. The models we list here are all compatible with our proposed SEPAL approach.

| Model | Relational operator $\phi$ |
|---|---|
| TransE (Bordes et al., 2013) | $\theta_h + \theta_r$ |
| MuRE (Balazevic et al., 2019) | $\theta_h \odot \rho_r - \theta_r$ |
| RotatE (Sun et al., 2019) | $\theta_h \odot \theta_r$ |
| QuatE (Zhang et al., 2019) | $\theta_h \otimes \theta_r$ |
| DistMult (Yang et al., 2014) | $\theta_h \odot \theta_r$ |
| ComplEX (Trouillon et al., 2016) | $\theta_h \odot \theta_r$ |
| TuckER (Balažević et al., 2019) | $\mathcal{W} \times_1 \theta_h \times_2 \theta_r$ |

**Embedding propagation** CompGCN (Vashishth et al., 2019) introduces the idea of propagating knowledge-graph embeddings using the relational operator $\phi$, but couples it with learnable weights and a non-linearity. REP (Wang et al., 2022) simplifies this framework by removing weight matrices and nonlinearities.

## 2.2 SCALING GRAPH ALGORITHMS

Various tricks help scale graph algorithms to the sizes we are interested in –millions of nodes.

**Graph partitioning** Scaling up computation on graph, for graph embedding or more generally, often relies on breaking down graphs in subgraphs. For this, the partitioning, clustering, and community-detection literatures are relevant. METIS (Karypis & Kumar, 1997), is a greedy node-merging algorithm heavily used to scale all types of graph algorithm. A variety of algorithms have also been developed to detect "communities", groups of nodes more connected together, often with applications on social networks: the *Label Propagation Algorithm* (LPA) (Raghavan et al., 2007), *spectral clustering* (SC) (Shi & Malik, 2000), Louvain method (Blondel et al., 2008), the *Leading Eigenvector* (LE) method (Newman, 2006), the Infomap method (Rosvall & Bergstrom, 2008), and the Leiden method (Traag et al., 2019) which guarantees connected communities.

**Local subsampling** Other forms of data reduction can help to scale graph algorithms (*e.g.* based on message passing). Algorithms may subsample neighborhoods, as GraphSAGE (Hamilton et al.,

2017) that selects a fixed number of neighbors for each node on each layer, or MariusGNN (Waleffe et al., 2023) that uses an optimized data structure for neighbor sampling and GNN aggregation. Cluster-GCN (Chiang et al., 2019) restricts the neighborhood search within clusters, obtained by classic clustering algorithms, to improve computational efficiency on graphs with a community structure. GraphSAINT (Zeng et al., 2019) creates overlapping subgraphs through random walks.

**Multi-level techniques.** Multi-level approaches, such as HARP (Chen et al., 2018), GraphZoom (Deng et al., 2019) or MILE (Liang et al., 2021), coarsen the graph, compute embeddings on the obtained smaller graph, and project them back to the original graph.

### 2.3 SCALING KNOWLEDGE-GRAPH EMBEDDING

The multi-relational aspect of knowledge graphs, captured *e.g.* in the $\phi$ detailed above, calls for scaling tailored methods.

**Parallel training.** Many approaches speed up triple-level stochastic solvers by distributing training across multiple workers, starting from the seminal PyTorch-BigGraph (PBG) (Lerer et al., 2019). The challenge is then to limit overheads and communication costs, as the embeddings of the relations are global trainable parameters, and thus require moving data between workers. For this, DGL-KE (Zheng et al., 2020) reduces data movement by using sparse relation embeddings and a min-cut-based graph partitioning algorithm (Karypis & Kumar, 1997, METIS) to distribute the triples across workers. HET-KG (Dong et al., 2022) further optimizes distributed training by preserving a copy of the few most frequently used embeddings on each worker, to reduce communication costs. These 'hot-embeddings' are periodically synchronized to minimize inconsistency. SMORE (Ren et al., 2022) leverages asynchronous scheduling to overlap CPU-based data sampling, with GPU-based embedding computations. Algorithmically, it contributes a bidirectional rejection sampling strategy to generate the negatives at a very low cost. GraphVite (Zhu et al., 2019) accelerates SGNS for graph embedding by both parallelizing random walk sampling on multiple CPUs, and negative sampling on multiple GPUs. Finally, Marius (Mohoney et al., 2021) optimizes data movement with 1) a data flow architecture that maximizes resource utilization of the entire memory hierarchy, including disk, CPU, and GPU memory, 2) Partition caching and a buffer-aware data ordering to minimize disk IO.

**Bags of entities.** Other attempts to scale knowledge graphs include StarSpace (Wu et al., 2018), that models some entities as bags of other entities rather than giving them explicit embeddings, or NodePiece (Galkin et al., 2021) that embeds a subset of entities called *anchors*, and learns an aggregation function to compute embeddings for all the other entities.

## 3 SEPAL: EXPANDING FROM A CORE SUBGRAPH

The work on scaling knowledge-graph embedding has mainly focused on efficient parallel computing to speed up stochastic optimization. We introduce a complementary approach, SEPAL, which changes how the embeddings are computed, avoiding much of the optimization cost. To extract rich node features from very large knowledge graphs, SEPAL allocates more computation time to the more frequent entities. To that end, SEPAL proceeds in two steps (Figure 1):

1. compute connected overlapping subgraphs that cover the full graph;

2. propagate the embeddings from the core to the outer subgraphs, with a message-passing strategy preserving the relational geometry.

SEPAL's key idea is to propagate embeddings to regions of the graph where they have not been computed yet, departing from embedding propagation methods (Vashishth et al., 2019; Wang et al., 2022) that use propagation as a post-processing to smooth pre-trained knowledge-graph embeddings. SEPAL is compatible with any embedding model whose scoring function has the form given by Equation 1, some examples of which are provided in Table 1.

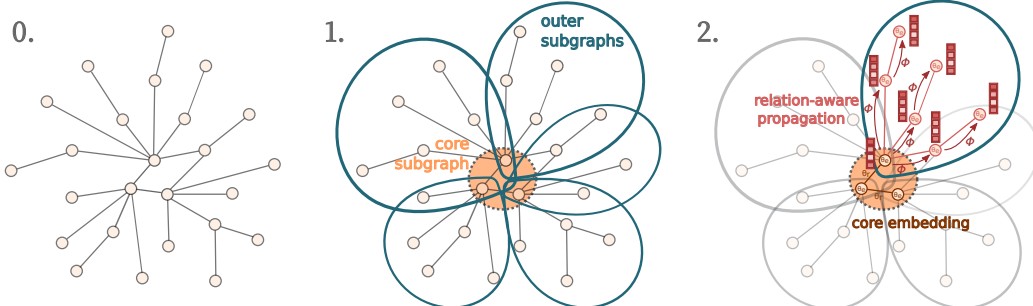

Figure 1: **SEPAL's general framework.** An input knowledge graph (*0.*) is first subdivided into BLOCS (*1.*). The core subgraph is then embedded, and the resulting embeddings are propagated to the outer subgraphs successively (*2.*).

## 3.1 SPLITTING LARGE GRAPHS WITH BLOCS

Breaking up the graph into subgraphs is key to scaling up our approach memory-wise. Specifically, we seek a set of subgraphs that altogether cover the full graph but are individually small enough to fit on GPUs, to enable the subsequent GPU-based message passing.

**Core subgraph.** SEPAL first defines the *core* of a knowledge graph as the subgraph induced by its most central entities. To build it, SEPAL focuses on entities, selecting the top $\eta\%$ entities by degree and keeping the largest connected component of the induced subgraph. The parameter $\eta$ is chosen large enough to ensure that the core subgraph contains all the relation types present in the graph (typically $\eta$ varies between 2 and 5%).

**Outer subgraphs.** The next class of subgraphs that we generate –the *outer* subgraphs– aim at covering the rest of the graph. The purpose of these subgraphs demands the following requirements:

**R1: connected** the subgraphs must be connected, to propagate the embeddings
**R2: bounded size** the subgraphs must have bounded sizes, to fit their embedding in GPU memory
**R3: coverage** the union of the subgraphs must be the full graph, to embed every entities
**R4: scalability** extraction must run with available computing resources, in particular memory

Extracting such subgraphs is challenging on large knowledge graphs. These are scale-free graphs with millions of nodes exhibit no well-defined clusters (Leskovec et al., 2009) and pose difficulties to existing partitioning algorithms. For instance, algorithms based on propagation, eigenvalues, or power iterations of the adjacency matrix (Raghavan et al., 2007; Shi & Malik, 2000; Newman, 2006) struggle with the presence of extremely high-degree nodes that make the adjacency matrix ill-conditioned. To satisfy our requirements despite these challenge, we allow subgraphs to overlap.

We contribute BLOCS, an algorithm designed to break large graphs into Balanced Local Overlapping Connected Subgraphs. The name summarizes the goals: 1) **Balanced**: BLOCS produces subgraphs of comparable sizes. $m$, the upper bound for subgraph sizes, is a hyperparameter. 2) **Local**: the subgraphs have small diameters. The essence of BLOCS minimizes the intrasubgraph mean shortest path length using a diffusion step. This locality property is important for the efficiency of SEPAL's propagation phase, as it intuitively reduces the number of propagation iterations needed to converge to the global embedding structure. 3) **Overlapping**: a given node can belong to several subgraphs. This is beneficial to our purpose because it facilitates information transfer between the different subgraphs during the propagation. 4) **Connected**: all generated subgraphs are connected.

BLOCS uses three base mechanisms to grow the subgraphs: **diffuse** (add all neighboring entities to the current subgraph), **merge** (merge two overlapping subgraphs) and **dilate** (add all unassigned neighboring entities to the current subgraph). There are two different regimes during the generation of subgraphs. First, few entities are assigned, and the computationally effective diffusion quickly covers a large part of the graph, especially entities that are close to high-degree nodes. However, once these close entities have been assigned, the effectiveness of diffusion drops because it strug-

---

**Algorithm 1** BLOCS algorithm

---

1: **Input:** Graph $\mathcal{G} = (V, E)$ with nodes $V$ and edges $E$, hyperparameters $h$ and $m$
2: **Output:** List of overlapping connected subgraphs
3: $\mathbb{S} \leftarrow \emptyset$                                                  ▷ list of subgraphs
4: $U \leftarrow V$                                              ▷ set of unassigned nodes
5: **Step 1: Create subgraphs from high-degree nodes' neighbors**
6: **for** each node $v \in V$ **do**
7:     **if** $deg(v) > 0.2\,m$ **then**
8:         $\mathbb{S}, U \leftarrow$ EQUALLYSPLITNEIGHBORSINSUBGRAPHS($v$, max_size $= 0.2\,m$)
9:     **end if**
10: **end for**
11: **Step 2: Assign nodes to subgraphs by diffusion**
12: **while** $|U| > (1-h)|V|$ **do**
13:     $k \leftarrow 0$   ;   $\mathcal{S}_0 \leftarrow \{\arg\max_{v \in U} deg(v)\}$     ▷ start with unassigned node $v$ with highest degree
14:     **while** $|\mathcal{S}_k| < 0.8\,m$ **do**
15:         $\mathcal{S}_{k+1} \leftarrow$ DIFFUSE($\mathcal{S}_k$)   ;   $k \leftarrow k+1$
16:     **end while**
17:     Append $\mathcal{S}_{k-1}$ to $\mathbb{S}$, and update $U$     ▷ $\mathcal{S}_{k-1}$ is the last subgraph smaller than $0.8\,m$
18: **end while**
19: **Step 3: Merge small overlapping subgraphs**
20: $\mathbb{S}, U \leftarrow$ MERGESMALLSUBGRAPHS($\mathbb{S}$, min_size $= m/2$)
21: **Step 4: Dilation and diffusion until all entities are assigned**
22: $p \leftarrow 0$
23: **while** $|U| > 0$ **do**
24:     **if** 5 divides $p$ and $p > 0$ **then**▷ create new subgraphs by diffusion every 5 rounds, to tackle long chains
25:         **repeat** 10 **times**
26:             $k \leftarrow 0$   ;   $\mathcal{S}_0 \leftarrow \{\arg\max_{v \in U} deg(v)\}$
27:             **while** $|\mathcal{S}_k| < 0.8\,m$ **do**
28:                 $\mathcal{S}_{k+1} \leftarrow$ DIFFUSE($\mathcal{S}_k$)   ;   $k \leftarrow k+1$
29:             **end while**
30:             Append $\mathcal{S}_{k-1}$ to $\mathbb{S}$, and update $U$     ▷ $\mathcal{S}_{k-1}$ is the last subgraph smaller than $0.8\,m$
31:     **end**
32:     **end if**
33:     $\mathbb{S} \leftarrow$ DILATE($\mathbb{S}$)   ;   $p \leftarrow p+1$
34: **end while**
35: **Step 5: Merge small overlapping subgraphs again**
36: $\mathbb{S}, U \leftarrow$ MERGESMALLSUBGRAPHS($\mathbb{S}$, min_size $= m/2$)
37: **Step 6: Split subgraphs larger than $m$**
38: $\mathbb{S}, U \leftarrow$ SPLITLARGESUBGRAPHS($\mathbb{S}$, max_size $= m$)
39: **Return:** $\mathbb{S}$, the set of overlapping subgraphs covering the graph $\mathcal{G}$

---

gles to reach entities farther away. For this reason, BLOCS switches from diffusion to dilation once the proportion of assigned entities reaches a certain threshold $h$ (chosen $\approx .6$, depending on the dataset). By adding only unassigned neighbors to subgraphs, dilation drives subgraph growth towards unassigned distant entities. However, the presence of long chains can drastically slow this regime, because they make it add entities one by one. Some knowledge graphs have long chains, for instance YAGO4.5 (see Mean Shortest Path Length in Appendix B). To tackle them, BLOCS switches back to diffusion for a few steps, with seeds taken inside the long chains.

The design of BLOCS makes it work faster on graphs that have small diameters, where most entities can be reached during the diffusion regime and very few dilation steps are required.

### 3.2 SEPARATING CORE OPTIMIZATION AND OUTER PROPAGATION

**Embedding the core**   Once the core subgraph is defined, embeddings for the relations and core entities are trained on GPU using any triple-based embedding model that fits with our framework. We add the inverse relations, ensuring connectedness for the subsequent propagation step.

**Relation-aware embedding propagation by message passing**   The key to SEPAL's computational efficiency is not requiring any gradient descent for the embeddings of the outer entities. Instead, the final step involves an embedding propagation that is consistent with the KGE model

(multiplication for DistMult, addition for TransE, ...) and preserves the relational geometry of the embedding space. To do so, SEPAL leverages the entity-relation composition function $\phi$ used by the knowledge-graph embedding model, and the embeddings of the relations $\theta_r$ trained on the core subgraph. From Equation 1 one can derive, for a given triple $(h, r, t)$, the closed-form expression of the tail embedding that maximizes the scoring function $\arg\max_{\theta_t} f(h, r, t) = \phi(\theta_h, \theta_r)$. SEPAL uses this property to compute the embeddings of the outer entities, by propagating from core entities with message passing.

First, the embeddings are initialized with $\quad \theta_u^{(0)} = \begin{cases} \theta_u, & \text{if entity } u \text{ belongs the core subgraph,} \\ \mathbf{0}, & \text{otherwise.} \end{cases}$

Then, each outer subgraph is merged with the core subgraph, and SEPAL loads its embeddings on GPU and performs $K$ steps of propagation, satisfying the following message-passing equations:

$$m_{u,v}^{(t+1)} = \sum_{(u,r,v) \in \mathcal{K}} \phi(\theta_v^{(t)}, \theta_r) \qquad \text{(message; } \phi \text{ is given by Table 1)}$$

$$a_u^{(t+1)} = \sum_{v \in \mathcal{N}(u)} m_{u,v}^{(t+1)} \qquad \text{(aggregation)}$$

$$\theta_u^{(t+1)} = \text{NORMALIZE}(\theta_u^{(t)} + a_u^{(t+1)}) \qquad \text{(update)}$$

where $\mathcal{N}(u)$ denotes the set of neighbors of entity $u$, and $\mathcal{K}$ the set of positive triples of the graph. During updates, $\ell_2$ normalization projects embeddings on the unit sphere. This accelerates convergence by canceling the effect of neighbors that still have zero embeddings. Normalizing embeddings is a common practice of knowledge-graph embedding models (Bordes et al., 2013; Yang et al., 2014), and SEPAL acts consistently. During propagation, the core embeddings remain frozen.

## 4 EXPERIMENTAL STUDY

### 4.1 KNOWLEDGE GRAPH DATASETS

To compare large knowledge graphs of different sizes, we use three different generations of YAGO: YAGO3 (Mahdisoltani et al., 2014), YAGO4 (Pellissier Tanon et al., 2020), and YAGO4.5 (Suchanek et al., 2023). We expand YAGO4 and YAGO4.5 into a larger version that also contains the taxonomy, i.e., types and classes –which algorithms will treat as entities– and their relations. We discard numerical attributes and keep only the largest connected component (Appendix B). To perform an ablation study of SEPAL without BLOCS for which we need smaller datasets, we also introduce Mini YAGO3, a subset of YAGO3 built by extracting the 5% most frequent entities.

### 4.2 EVALUATING NODE FEATURES ON DOWNSTREAM REGRESSION TASKS

We evaluate the embeddings as node features, used to facilitate learning in downstream tasks (Grover & Leskovec, 2016; Cvetkov-Iliev et al., 2023; Robinson et al., 2024). This task enables to compare the value of knowledge graphs of different sizes. Indeed, for a user, a suboptimal embedding of a larger knowledge graph may be more interesting than a high-quality embedding of a smaller knowledge graph because the larger graph brings information on more entities. We benchmark 4 downstream regression tasks (adapted from Cvetkov-Iliev et al., 2023): Movie revenues, US accidents, US elections, and housing prices (see Appendix C).

Figure 2 gives the prediction performance on the downstream tasks. SEPAL not only scales well to very large graphs (computing times markedly smaller than Pytorch-BigGraph), but also create more valuable node features for downstream tasks. Larger knowledge graphs do bring value, as they cover more entities of the downstream task (Table 8). The good performance of FastRP provides insights on why SEPAL improves on the performance of its base models: as SEPAL's second step, it is based on iterating graph propagations, which structures the embeddings.

### 4.3 EVALUATING BLOCS

**Compared to other partitioning algorithms** We first compare BLOCS to other graph partitioning, clustering, and community detection methods. Table 2 reports empirical evaluation on our four

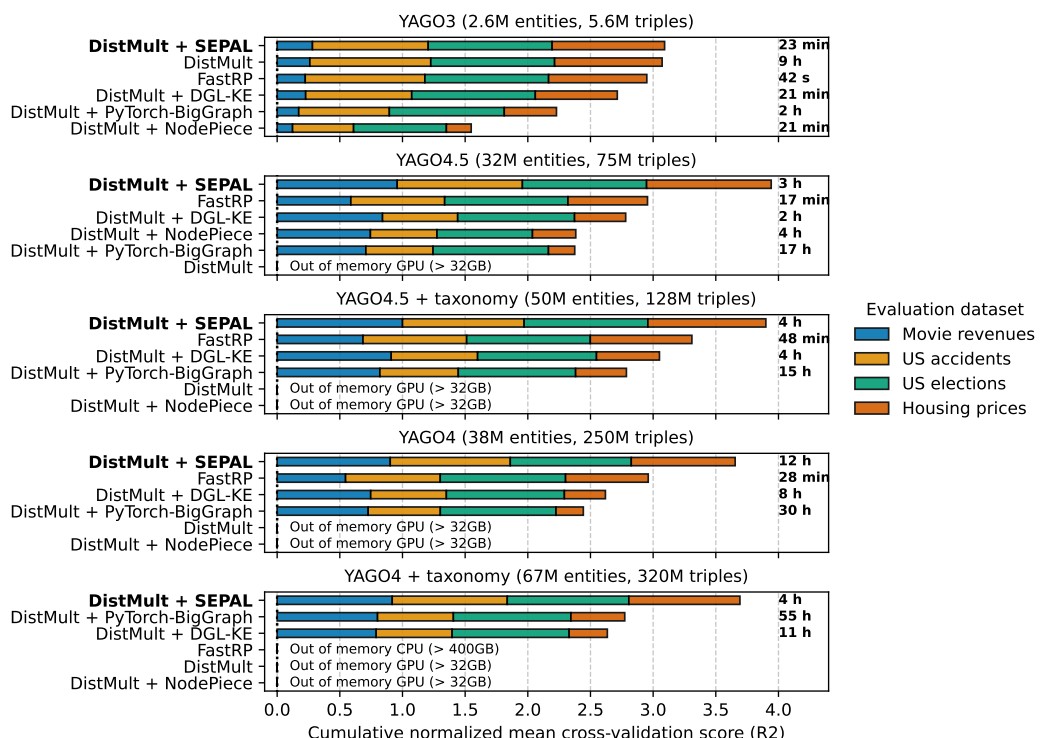

Figure 2: **Performance on downstream tasks**: normalized score (*ie* for an evaluation dataset, 1 corresponds to the best R2 score across all models). SEPAL, PyTorch-BigGraph, DGL-KE, and NodePiece use DistMult as base model.

knowledge graphs. BLOCS and METIS are the only approaches that scale to the largest knowledge graphs. Others fail due to excessive runtimes –our limit was set to $10^4$ seconds. Compared to METIS, BLOCS is more efficient in terms of RAM usage while having similar computation times. Experimental results also show that classic partitioning methods fail to meet the connectedness and size requirements. Indeed, knowledge graphs are prone to yield disconnected partitions due to their scale-free nature: they contain very high degree nodes. Such a node is hard to allocate to a single subgraph, and subgraphs without it often explode in multiple connected components. Our choice of overlapping subgraphs avoids this problem.

**BLOCS inside SEPAL: ablation study** Here, we study the effect of removing BLOCS from our proposed method. On smaller knowledge graphs, SEPAL can be used with a simple core subgraph extraction and embedding followed by the embedding propagation. This ablation reveals the impact of BLOCS on the model's performance. Figure 3 shows that adding BLOCS to the pipeline on graphs that would not need it (because they are small enough for all the embeddings to fit in GPU memory) does not alter performance, showing that BLOCS enables efficient embedding propagation through message-passing. Additionally, BLOCS brings scalability. By tuning the maximum subgraph size $m$ hyperparameter, one can move the blue points horizontally on Figure 3 and choose a value within the GPU constraints. There is a trade-off between decreasing GPU RAM usage (*i.e.* moving the blue points to the left) and increasing execution time, as this requires more data movement between CPU and GPU.

### 4.4 OUTER PERFORMANCE

Table 3 shows the scores of the different methods for the entities of the outer subgraphs. This experiment demonstrates the effectiveness of SEPAL's propagation step, as all the SEPAL embeddings evaluated here have been computed through propagation. Results show that SEPAL's propagated embeddings compare favorably to all the baselines on four of the five knowledge graphs.

Table 2: **Scalability and performance of clustering methods**: whether each method experimentally complies with requirements R1 and R2, as well as computation time and RAM usage. Table 5 in appendix gives results on Mini YAGO3, showing that spectral clustering fails to meet R1 and R2, and uses much RAM.

As the graph size increases, an increasing number of algorithms do not run with available resources, and are not displayed in the table. Appendix F.2 provides more details.

| | R1 (connected) | R2 (bounded size) | Time | RAM usage |
|---|:---:|:---:|---|---|
| **a. YAGO3** | | | | |
| **BLOCS** | ✓ | ✓ | 98.7 s | 2.68 GB |
| METIS | ✗ | ✓ | 50.7 s | 5.43 GB |
| LE | ✓ | ✗ | 41.0 s | 2.65 GB |
| Leiden | ✓ | ✗ | 101 s | 2.37 GB |
| Louvain | ✗ | ✗ | 101 s | 3.11 GB |
| Infomap | ✗ | ✗ | 1580 s | 6.00 GB |
| LPA | ✗ | ✗ | 607 s | 2.24 GB |
| **b. YAGO4.5** | | | | |
| **BLOCS** | ✓ | ✓ | 53.2 min | 25.1 GB |
| METIS | ✗ | ✓ | 16.0 min | 68.0 GB |
| LE | ✗ | ✗ | 127 min | 54.3 GB |
| Leiden | ✓ | ✗ | 39.0 min | 54.0 GB |
| Louvain | ✗ | ✗ | 163 min | 54.5 GB |
| **c. YAGO4.5 + taxonomy** | | | | |
| **BLOCS** | ✓ | ✓ | 22.3 min | 47.5 GB |
| METIS | ✗ | ✓ | 35.4 min | 120 GB |
| **d. YAGO4** | | | | |
| **BLOCS** | ✓ | ✓ | 72.3 min | 63.2 GB |
| METIS | ✗ | ✓ | 65.2 min | 209 GB |
| LE | ✓ | ✗ | 33.9 min | 157 GB |
| **e. YAGO4 + taxonomy** | | | | |
| **BLOCS** | ✓ | ✓ | 22.7 min | 119 GB |

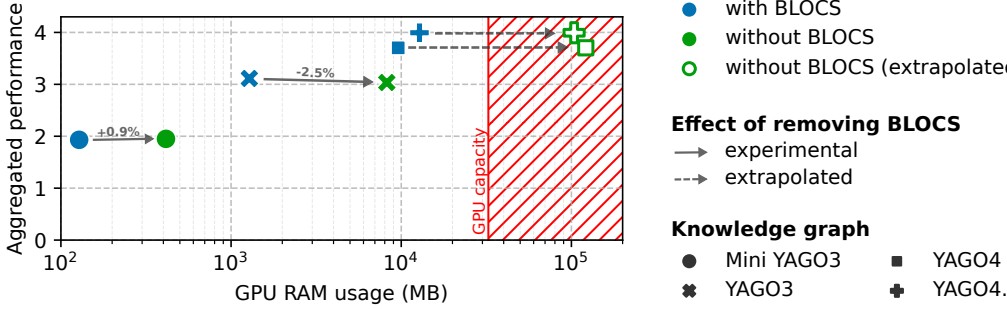

Figure 3: **Ablation study: BLOCS scales SEPAL memory-wise.** Normalized R2 scores (same as Figure 2) aggregated across evaluation datasets (movie revenues, US accidents, US elections, housing prices) for SEPAL with and without BLOCS are plotted against GPU RAM usage (see Appendix F.1). The relative performance variation when removing BLOCS is indicated above the arrows. BLOCS preserves performance for a given knowledge graph while drastically reducing memory pressure on GPU RAM. Without BLOCS, the GPU runs out of memory for YAGO4 and YAGO4.5.

## 5 DISCUSSION AND CONCLUSION

**Modern embeddings on modern knowledge graphs with modest hardware**   SEPAL reconciles the evergrowing size of knowledge graphs with the evergrowing sophistication of knowledge-graph embeddings. Indeed, it brings marked computational-performance benefits when embedding large knowledge graphs: multiple-fold decreased train times and bounded memory usage. It achieves this speed-up without requiring heavy engineering, such as distributed computing, and can easily be adapted to most knowledge-graph embedding methods. SEPAL improves the quality of the gener-

Table 3: Normalized mean cross-validation score (R2) in outer graphs. Best in bold, second underlined.

| | Housing prices | Movie revenues | US accidents | US elections | Average score |
|---|---|---|---|---|---|
| **a. YAGO3** (2.4M outer entities) | | | | | |
| **DistMult + SEPAL** | **1.000** | 0.228 | 0.853 | 0.952 | **0.758** |
| DistMult | 0.842 | 0.283 | 0.822 | **0.957** | 0.726 |
| PyTorch-BigGraph | 0.451 | 0.270 | 0.863 | 0.883 | 0.617 |
| DGL-KE | 0.685 | **0.393** | **0.991** | 0.950 | 0.755 |
| NodePiece | 0.149 | 0.078 | 0.158 | 0.806 | 0.298 |
| FastRP | 0.902 | 0.204 | 0.799 | 0.941 | 0.712 |
| **b. YAGO4.5** (31M outer entities) | | | | | |
| **DistMult + SEPAL** | **0.896** | 0.473 | **0.757** | 0.984 | **0.778** |
| PyTorch-BigGraph | 0.201 | 0.778 | 0.607 | 0.889 | 0.619 |
| DGL-KE | 0.408 | **0.931** | 0.689 | 0.896 | 0.731 |
| NodePiece | 0.178 | 0.339 | 0.116 | 0.985 | 0.404 |
| FastRP | 0.461 | 0.135 | 0.396 | **0.994** | 0.497 |
| **c. YAGO4.5 + taxonomy** (48M outer entities) | | | | | |
| **DistMult + SEPAL** | **0.857** | 0.595 | 0.741 | 0.952 | 0.786 |
| PyTorch-BigGraph | 0.402 | 0.904 | 0.720 | 0.902 | 0.732 |
| DGL-KE | 0.498 | **1.000** | **0.798** | 0.914 | **0.803** |
| FastRP | 0.747 | 0.422 | 0.532 | **1.000** | 0.675 |
| **d. YAGO4** (37M outer entities) | | | | | |
| **DistMult + SEPAL** | **0.826** | **0.999** | **1.000** | 0.955 | **0.945** |
| PyTorch-BigGraph | 0.211 | 0.793 | 0.654 | 0.888 | 0.636 |
| DGL-KE | 0.321 | 0.826 | 0.692 | 0.905 | 0.686 |
| FastRP | 0.684 | 0.587 | 0.672 | **0.960** | 0.726 |
| **e. YAGO4 + taxonomy** (66M outer entities) | | | | | |
| **DistMult + SEPAL** | **0.906** | **0.996** | **0.919** | **0.956** | **0.944** |
| PyTorch-BigGraph | 0.422 | 0.886 | 0.691 | 0.902 | 0.725 |
| DGL-KE | 0.299 | 0.862 | 0.701 | 0.898 | 0.690 |

ated node features when used for data enrichment in external (downstream) tasks, a setting that can strongly benefit from pre-training embeddings on knowledge bases as large as possible.

Insights brought by our experiments go further than SEPAL. First, the method successfully exploits the asymmetry of information between "central" entities and more peripheral ones. Power-law distributions are indeed present on many types of objects, from words (Piantadosi, 2014) to geographical entities (Giesen & Südekum, 2011) and should probably be exploited for general-knowledge representations such as knowledge-graph embeddings. Second, and related, breaking up large knowledge graphs in communities is surprisingly difficult: some entities just belong in many (all?) communities, and others are really hard to reach. Our BLOCS algorithm can be useful for other knowledge-graph engineering tasks, such as scaling message-passing algorithms or simply generating partitions. Finally, the embedding propagation in SEPAL appears powerful and we conjecture it will benefit further approaches. First, it can be combined with much of the prior art to scale knowledge-based embedding. Second, it seems a natural solution for link prediction semi-inductive settings: link prediction on nodes newly connected to the graph (Ali et al., 2021a; Galkin et al., 2021), that thus could be easily embedded by propagation. Finally, embedding propagation could naturally adapt to continual learning settings (Van de Ven & Tolias, 2019; Hadsell et al., 2020; Biswas et al., 2023)

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

## A ADDITIONAL RESULTS

### A.1 EXTENDED RESULTS: MORE EMBEDDING MODELS, MORE DATASETS

Figure 4 extends the results of Figure 2 by adding TransE and RotatE, alone and combined with SEPAL, as well as the Freebase dataset. This demonstrates that:

1. SEPAL scales to knowledge graphs up to 85M entities.

2. SEPAL adapts to embedding models other than DistMult, such as TransE and RotatE, and even improves on its base model.

For a fair comparison, we ran RotatE with embedding dimension $d = 50$, as it outputs complex embeddings having twice as many parameters. For other models, we use $d = 100$.

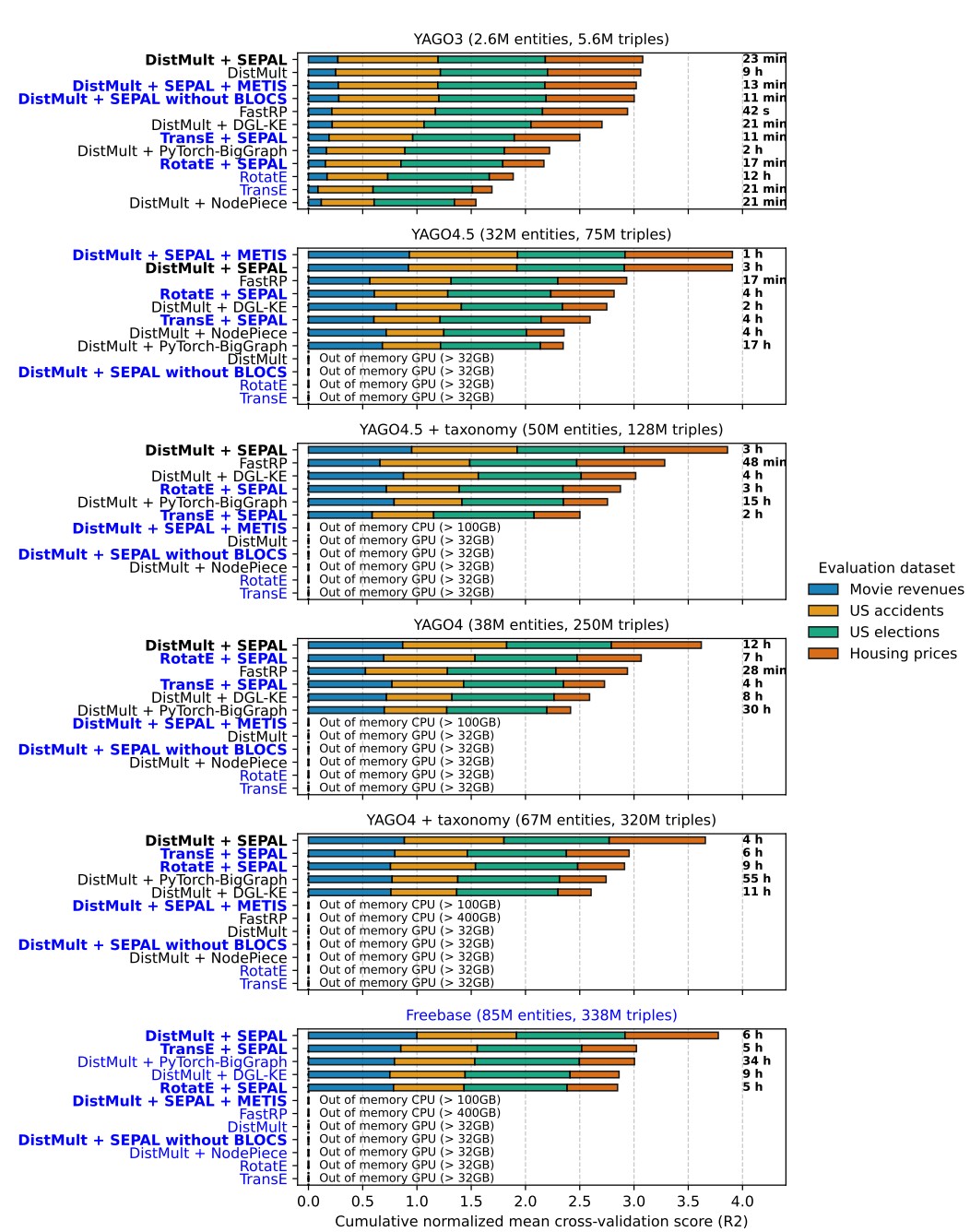

Figure 4: **Performance on downstream tasks.**

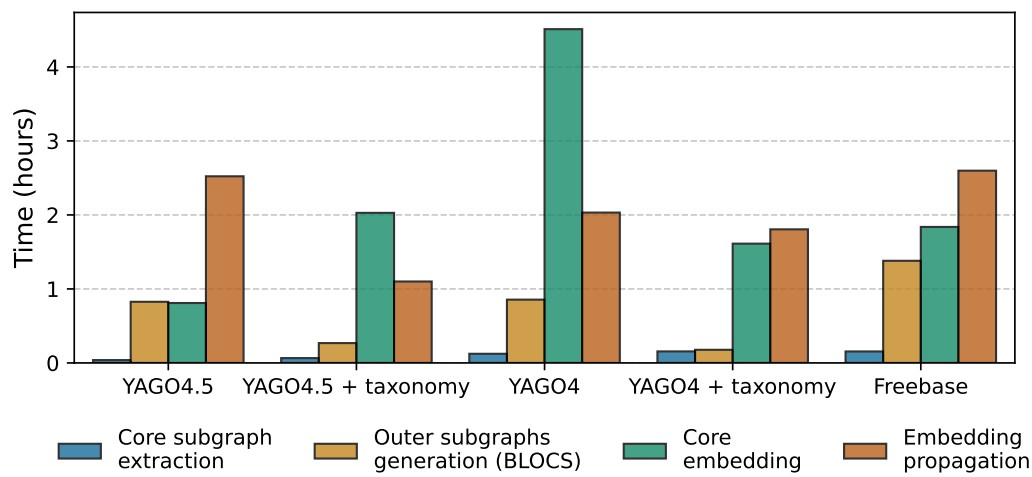

Figure 5: SEPAL's execution time breakdown.

## A.2 EXECUTION TIME BREAKDOWN

Here, we present the contribution of each part of the pipeline to the total execution time. Specifically, we break down our method into four parts:

1. core subgraph extraction;
2. outer subgraphs generation (BLOCS);
3. core embedding;
4. embedding propagation.

Figure 5 shows the execution time of the different components of SEPAL. It includes our five largest knowledge graphs, for which the execution times have the same order of magnitude. The results reveal that most of the execution time is due to the core embedding and embedding propagation phases, while the core extraction time is negligible.

Three key factors influence SEPAL's execution time during the three main steps of the pipeline:

1. **The core subgraph size**: the more triples in the core subgraph, the longer the core embedding. This explains the wide disparities between the core embedding times on Figure 5, despite all the core subgraphs having roughly the same number of entities: YAGO4 core subgraph is more dense (33M triples), compared to YAGO4.5 (7M triples) for instance. Of course, the core embedding time also depends on hyperparameters such as the number of training epochs.

2. **The diameter of the knowledge graph**: graphs with large diameters call for more dilation steps during BLOCS' subgraph generation, and dilation is more costly than diffusion because it requires checking node assignments. This explains why adding the taxonomies to YAGO4 and YAGO4.5 drastically reduces the time required to run BLOCS, as shown on Figure 5.

3. **The total number $N$ of entities in the graph**: this number determines the size of the embedding matrix. The communication cost of moving embedding matrices from CPU to GPU, and vice versa, accounts for most of the propagation time, and increases with $N$. It also increases with the amount of overlap between the outer subgraphs produced by BLOCS, explaining the differences in propagation time between YAGO4.5 and YAGO4.5 + taxonomy for instance.

Interestingly, the number of propagation steps $K$ as little impact on the embedding propagation time. The reason for this is that much of this time stems from the communication cost of loading the embeddings onto the GPU, and not from performing the propagation itself.

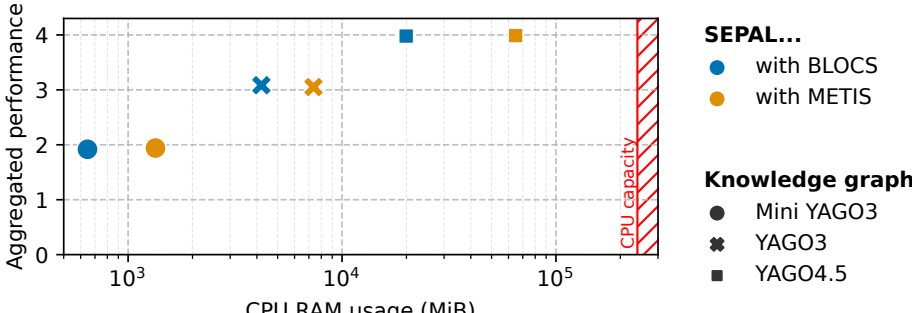

Figure 6: **Ablation study: replacing BLOCS with METIS.**. Normalized R2 scores (same as Figure 2) aggregated across evaluation datasets (movie revenues, US accidents, US elections, housing prices) for SEPAL with BLOCS and METIS are plotted against CPU RAM usage. BLOCS necessitates significantly less memory than METIS. We were not able to run SEPAL + METIS on knowledge graphs larger than YAGO4.5, hitting CPU RAM limits.

## A.3    BLOCS CANNOT BE REPLACED WITH METIS

To demonstrate the benefits of BLOCS over existing methods, we try to replace BLOCS with METIS in our framework. The results are presented in Figure 6.

Two important points differentiate these methods:

1. Contrary to BLOCS, METIS outputs disconnected partitions (see Table 2). Given the structure of SEPAL, this results in zero-embeddings for entities not belonging to the core connected component at propagation time. Interestingly, the presence of zero-embeddings affects downstream scores very little, likely because most downstream entities belong to the core connected component and are thus not impacted by this.

2. METIS does not scale as well as BLOCS in terms of CPU memory. Therefore BLOCS remains an indispensable ingredient for graphs larger than YAGO4.5 (32M entities).

## A.4    COMPARISON WITH NODEPIECE

SEPAL shares with NodePiece the fact that it embeds a subset of entities. Parallels can be drawn between: a) the anchors of NodePiece and the core entities of SEPAL; b) the encoder function of NodePiece and the embedding propagation of SEPAL. Yet, our approach differs from NodePiece in several ways.

**Neighborhood context handling.**    Both methods handle completely differently the neighborhood of entities. NodePiece tokenizes each node into a sequence of $k$ anchors and $m$ relation types, where $k$ and $m$ are fixed hyperparameters shared by all nodes. If the node degree is greater than $m$, NodePiece downsamples randomly the relation tokens, and if it is lower than $m$, [PAD] tokens are appended; both seem sub-optimal. In contrast, SEPAL accommodates any node degree and uses all the neighborhood information, thanks to the message-passing approach that handles the neighborhood context.

Additionally, NodePiece's tokenization relies on an expensive BFS anchor search, unsuitable for huge graphs. On our hardware, we could not run the vanilla NodePiece (PyKEEN implementation) on graphs bigger than Mini YAGO3 (129k entities). For YAGO3 and YAGO4.5, we had to run an ablated version where nodes are tokenized only from their relational context (i.e., $k = 0$, studied in the NodePiece paper with good results), to skip the anchor search step.

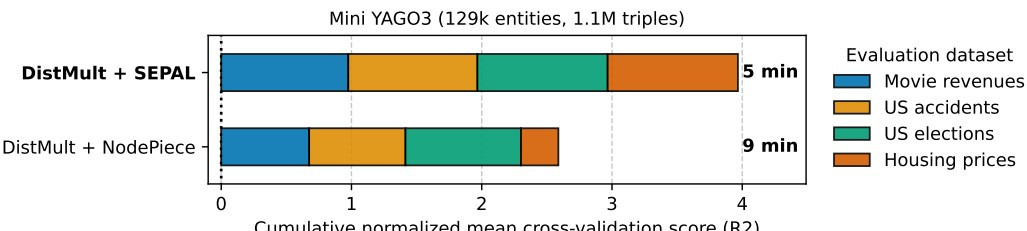

Figure 7: Comparing SEPAL with NodePiece on Mini YAGO3.

**Training procedure.** At train time, NodePiece goes through the full set of triples at each epoch to optimize both the anchors' embeddings and the encoder function parameters, necessitating many gradient computes and resulting in long training times for large graphs. On the contrary, SEPAL performs mini-batch gradient descent only on the triples of the core subgraph, which provides significant time savings. To illustrate this, Figure 7 compares the performance of SEPAL and vanilla NodePiece on Mini YAGO3, showing that SEPAL outperforms NodePiece on downstream tasks while being nearly two times quicker.

**Embedding propagation to non-anchor/non-core entities.** To propagate to non-anchor entities, NodePiece uses an encoder function (MLP or Transformer) that has no prior knowledge of the relational structure of the embedding space, and has to learn it through gradient descent. On the contrary, SEPAL leverages the model-specific relational structure to compute the outer embeddings with no further training needed.

## A.5 OTHER CORE SELECTION STRATEGIES

SEPAL selects the core entities based on degree. This is convenient for two reasons:

1. Degree is inexpensive to compute, ensuring the core extraction phase to be fast (see Figure 5);
2. It yields very dense core subgraphs. Indeed, while they contain $\eta\%$ of the entities of the full graph, they gather around $4\eta\%$ of all the triples (Table 4). This allows the training on the core to process a substantial portion of the graph triples, resulting in richer representations.

However, a problem is that some relation types may not be included in the core subgraph.

### A.5.1 RELATION-BASED CORE SELECTION STRATEGY

To deal with the relational coverage issue, we introduce another method for selecting the core subgraph. It is based on the relations: for each relation type, select the edges with the top $\eta\%$ degree (sum of degrees of head and tail) and keep the corresponding entities, then select the largest connected component of the induced subgraph.

This approach enhances relational coverage but, as a counterpart, yields a sparser core subgraph (Table 4). For YAGO4 and Freebase, the datasets on which the problem arises, switching from degree-selection to relation-selection increases the number of relation types in the core from 61 to 75 (over 76) on YAGO4, and from 5,363 to 14,266 (over 14,665) on Freebase. Some relation types are still missing because they were not present in the largest connected component of the induced subgraph.

Performance-wise, Figure 8 shows the impact of the core selection strategy. Simply using the degree appears to consistently give better than relation-based core selection results on downstream tasks. This is probably due to a richer core subgraph, containing more triples to train good core embeddings (Table 4). However, the sparser core resulting from the relation selection strategy also makes it faster, as there are fewer triples during core training.

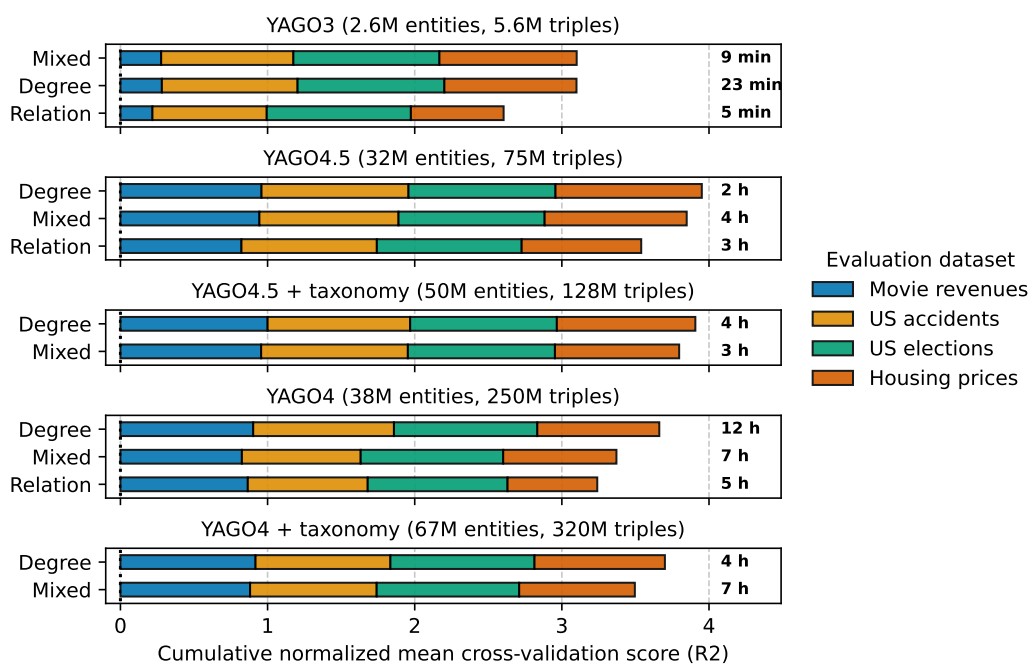

Figure 8: Performance of SEPAL+DistMult for our three core selection strategies: degree, relation, and mixed.

### A.5.2 MIXED CORE SELECTION STRATEGY

The relation-based core selection strategy still presents two significant drawbacks: it triggers a performance drop compared to the degree-based approach, and some relations are still missing in the core. For this reason, we introduce a "mixed" core selection strategy, aiming to get the best of both worlds.

The mixed core selection strategy proceeds in four main steps:

1. **Degree selection**: Sample the nodes with the top $\eta_n$ degrees.

2. **Relation selection**: Sample the edges with the top $\eta_e$ degrees (sum of degrees of head and tail) for each relation type, and keep the corresponding entities.

3. **Merge**: Take the union of these two sets of entities.

4. **Reconnect**: If the induced subgraph has several connected components, add entities to make it connected to the core. This is done using a breadth-first search (BFS) with early stopping from the node with the highest degree of each given connected component (except the largest) to the largest connected component. For each connected component (except the largest), a path linking it to the largest connected component is added to the core subgraph.

This way, each relation type is guaranteed to belong to the core subgraph, by design. Table 4 shows that it is indeed the case experimentally, even for Freebase which has 14,665 relation types.

This method features two hyperparameters $\eta_n$ and $\eta_e$, the proportions for node and edge selections, controlling the size of its output subgraph. The values we used are provided in Table 4 for each dataset.

Regarding performance, Figure 8 reveals that the mixed strategy helps bridge the gap between the degree and relation selection strategies. It consistently performs better than the simple relation-based strategy and concedes little to the degree-based approach. Furthermore, one can adjust the values of $\eta_n$ and $\eta_e$ to control the trade-off between downstream performance and relation coverage.

Table 4: **Effect of core selection strategies**: Number of entities and triples inside the core subgraph and the proportion of the full graph they represent (in parentheses). $\eta_n$ and $\eta_e$ are the hyperparameters for nodes and edges, respectively. Column #Rel gives the number of relation types present in the core compared to the total number of relation types in the knowledge graph. We highlight in red the cases where some relations are missing. Column Time gives the measured computation time for core selection.

| | Strategy | #Rel | #Entities | #Triples | Time |
|---|---|---|---|---|---|
| **YAGO3** | Degree ($\eta_n = 5\%$) | 37/37 | 126 k (4.9%) | 1.0 M (18.5%) | 17 s |
| | Relation ($\eta_e = 2.5\%$) | 37/37 | 132 k (5.2%) | 565 k (10.1%) | 20 s |
| | Mixed ($\eta_n, \eta_e = 2.5\%, 1.5\%$) | 37/37 | 121 k (4.7%) | 733 k (13.1%) | 20 s |
| **YAGO4.5** | Degree ($\eta_n = 3\%$) | 62/62 | 932 k (2.9%) | 7.2 M (9.6%) | 2 min |
| | Relation ($\eta_e = 2\%$) | 62/62 | 700 k (2.2%) | 2.5 M (3.3%) | 5 min |
| | Mixed ($\eta_n, \eta_e = 1.5\%, 1\%$) | 62/62 | 1.1 M (3.3%) | 5.7 M (7.5%) | 6 min |
| **YAGO4** | Degree ($\eta_n = 3\%$) | 61/76 | 1.1 M (3.0%) | 33 M (13.4%) | 8 min |
| | Relation ($\eta_e = 1\%$) | 75/76 | 1.7 M (4.6%) | 20 M (8.2%) | 20 min |
| | Mixed ($\eta_n, \eta_e = 1.5\%, 0.5\%$) | 76/76 | 1.4 M (3.8%) | 28 M (11.1%) | 11 min |
| **YAGO4.5 + taxo** | Degree ($\eta_n = 3\%$) | 64/64 | 1.5 M (3.0%) | 13 M (9.9%) | 4 min |
| | Relation ($\eta_e = 1\%$) | 64/64 | 1.2 M (2.4%) | 3.6 M (2.8%) | - |
| | Mixed ($\eta_n, \eta_e = 1.5\%, 0.5\%$) | 64/64 | 1.2 M (2.5%) | 8.3 M (6.5%) | 5 min |
| **YAGO4 + taxo** | Degree ($\eta_n = 2\%$) | 64/78 | 1.3 M (2.0%) | 41 M (12.8%) | 9 min |
| | Relation ($\eta_e = 1\%$) | 78/78 | 2.1 M (3.3%) | 43 M (13.6%) | - |
| | Mixed ($\eta_n, \eta_e = 1\%, 0.5\%$) | 78/78 | 1.5 M (2.3%) | 32 M (10.1%) | 12 min |
| **Freebase** | Degree ($\eta_n = 2\%$) | 5,363/14,665 | 1.7 M (2.0%) | 15 M (4.4%) | 9 min |
| | Relation ($\eta_e = 1\%$) | 14,266/14,665 | 2.1 M (2.5%) | 11 M (3.3%) | - |
| | Mixed ($\eta_n, \eta_e = 1\%, 0.5\%$) | 14,665/14,665 | 1.9 M (2.3%) | 14 M (4.1%) | - |

Table 5: **Scalability and performance of clustering methods**: whether each method experimentally complies with requirements R1 and R2, as well as computation time and RAM usage on mini-yago3. This table extends table 2.

| **a. Mini YAGO3** | | | | |
|---|---|---|---|---|
| | R1 (connected) | R2 (bounded size) | Time | RAM usage |
| **BLOCS** | ✓ | ✓ | 1.8 s | 0.702 GB |
| METIS | ✗ | ✓ | 7.24 s | 1.35 GB |
| Infomap | ✗ | ✗ | 93.5 s | 1.31 GB |
| LE | ✗ | ✓ | 82.8 s | 1.24 GB |
| LPA | ✗ | ✗ | 7.05 s | 1.24 GB |
| Leiden | ✓ | ✗ | 4.13 s | 1.24 GB |
| Louvain | ✓ | ✗ | 7.92 s | 1.26 GB |
| Spectral Clustering | ✗ | ✗ | 52 s | 4.68 GB |

## A.6  CLUSTERING COMPARISON ON MINI YAGO3

Comparing subgraph extraction (clustering) methods on mini YAGO3 (Table 5) is interesting to add spectral clustering to the comparison, as it does not run on larger graphs.

Table 6: Additional statistics on the knowledge graph datasets used. MSPL stands for Mean Shortest Path Length. The LCC column gives the percentage of entities of the graph that are in the largest connected component.

| | Maximum degree | Average degree | MSPL | Diameter | Density | LCC |
|---|---|---|---|---|---|---|
| Mini YAGO3 | 65 711 | 12.6 | 3.3 | 11 | 1e-4 | 99.98% |
| YAGO3 | 934 599 | 4.0 | 4.2 | 23 | 2e-6 | 97.6% |
| YAGO4.5 | 6 434 121 | 4.5 | 5.0 | 502 | 1e-7 | 99.7% |
| YAGO4.5 + taxonomy | 6 434 122 | 5.0 | 4.0 | 5 | 1e-7 | 100% |
| YAGO4 | 8 606 980 | 12.9 | 4.5 | 28 | 3e-7 | 99.0% |
| YAGO4 + taxonomy | 32 127 569 | 9.4 | 3.4 | 6 | 1e-7 | 100% |
| Freebase | 10 754 238 | 4.9 | 4.7 | 100 | 6e-8 | 99.1% |

## B  STATISTICS ON KNOWLEDGE-GRAPH DATASETS

More statistics on the knowledge graph datasets are given in Table 6. Maximum and average degree figures highlight the scale-free nature of real-world knowledge graphs. The values for mean shortest path length (MSPL) and diameter (the diameter is the longest shortest path) are provided for the largest connected component (LCC). They are remarkably small, given the number of entities in the graphs. Contrary to other datasets, YAGO4.5 and Freebase contain 'long chains', which account for their larger diameters.

The density $D$ is the ratio between the number of edges $|E|$ and the maximum possible number of edges:

$$D = \frac{|E|}{|V|(|V| - 1)}$$

where $|V|$ denotes the number of nodes.

The LCC statistics show that for each knowledge graph, the largest connected component regroups almost all the entities.

## C  DOWNSTREAM TASKS

We use 4 downstream tasks adapted from Cvetkov-Iliev et al. (2023) who also investigate knowledge-graph embeddings to facilitate machine learning. The specific target values predicted for each dataset are the following:

**US elections** : predict the number of votes per party in US counties.

**Housing prices** : predict the average housing price in US cities.

**US accidents** : predict the number of accidents in US cities.

**Movie revenues** : predict the box-office revenues of movies.

For each dataset, we use scikit-learn's Histogram-based Gradient Boosting Regression Tree (Pedregosa et al., 2011) as regression estimator to predict the target value. The embeddings are the only features fed to the estimator, except for the US elections dataset for which we also include the political party. For embedding models outputting complex embeddings, such as RotatE, we simply concatenate real and imaginary parts before feeding them to the estimator.

The rows of the tables corresponding to entities not found in the knowledge graph are filled with NaNs as features for the estimator. This enables to compare the scores between different knowledge graphs (see Figure 2) to see the benefits obtained from embedding larger graphs.

The metric used is the R2 score, defined by:

$$R^2 = 1 - \frac{\sum_{i=1}^{N}(y_i - \hat{y}_i)^2}{\sum_{i=1}^{N}(y_i - \bar{y})^2}$$

Table 7: Number of rows in the downstream tables.

|  | US elections | Housing prices | US accidents | Movie revenues |
|---|---|---|---|---|
| Number of rows | 13 656 | 22 250 | 20 332 | 7 398 |

Table 8: Proportion of entities in the downstream tables that were matched to an entity of the knowledge graph.

|  | Mini YAGO3 | YAGO3 | YAGO4.5 | YAGO4.5 + taxonomy | YAGO4 | YAGO4 + taxonomy | Freebase |
|---|---|---|---|---|---|---|---|
| Housing prices | 19.1% | 92.2% | 99.7% | 99.7% | 99.1% | 99.8% | 85.6% |
| Movie revenues | 27.7% | 62.3% | 99.4% | 99.4% | 99.4% | 99.5% | 89.1% |
| US accidents | 21.1% | 87.3% | 97.6% | 97.6% | 96.8% | 98.0% | 78.6% |
| US elections | 74.1% | 99.3% | 99.0% | 99.0% | 99.1% | 99.1% | 98.3% |

where $N$ is the number of samples (rows) in the target table, $y_i$ is the target value of sample $i$, $\hat{y}_i$ is the value predicted by the estimator, and $\bar{y}$ is the mean value of the target variable.

To get the "*Cumulative normalized mean cross-validation score*" presented on Figure 2, we proceed as follows:

1. **Mean cross-validation score**: for each model[1] and evaluation dataset, R2 scores are averaged over 5 repeats of 5-fold cross-validations.

2. **Normalized**: for each evaluation dataset, we divide all the scores by the score of the best-performing model on this dataset. This makes the scores more comparable between the different evaluation datasets.

3. **Cumulative**: for each model, we sum its scores across every evaluation dataset. As there are 4 evaluation datasets, the highest possible score for a model is 4. Getting a score of 4 means that the model beats every model on every evaluation dataset.

Table 7 gives an overview of the sizes of the downstream tasks, and table 8 gives the proportion of entities in these tables that are described in the different knowledge graphs.

# D SEPAL HYPERPARAMETERS

## D.1 LIST HYPERPARAMETERS FOR SEPAL'S

Here, we list the hyperparameters for SEPAL. Table 9 gives the values of those who depend on the dataset.

- **Proportion of core nodes** $\eta$: the idea is to select it large enough to ensure good core embeddings, but not too large so that core embeddings fit in the GPU memory. Figure 9 shows the experimental effect of varying this parameter;

- **Stopping diffusion threshold** $h$: it is probably the hardest hyperparameter to tune, as it depends on the graph structure. Tuning is done empirically by monitoring the proportion of unassigned entities during the BLOCS algorithm. $h$ is chosen equal to the proportion that starts to stagnate during BLOCS' diffusion regime. A bad choice of $h$ can make BLOCS intractable;

- **Number of propagation steps** $K$: it is chosen high enough to ensure reaching the remote entities (otherwise, they will have zeros as embeddings). Taking $K$ equal to the graph's diameter guarantees that this condition is fulfilled. However, for graphs with long chains, this may slow down SEPAL too much. In practice, choosing $K$ at 2–3 times the Mean Shortest Path Length (MSPL) usually embeds most entities effectively;

---

[1]A "*model*" is the combination of a method (*e.g.* DistMult, DGL-KE, etc.) and a knowledge graph on which it is trained.

Table 9: SEPAL dataset-specific hyperparameters used.

| | Mini YAGO3 | YAGO3 | YAGO4.5 | YAGO4.5 + taxonomy | YAGO4 | YAGO4 + taxonomy | Freebase |
|---|---|---|---|---|---|---|---|
| $\eta$ | 5% | 5% | 3% | 3% | 3% | 2% | 2% |
| $h$ | 0.8 | 0.77 | 0.6 | 0.8 | 0.55 | 0.8 | 0.55 |
| $K$ | 5 | 15 | 50 | 20 | 20 | 20 | 15 |
| $n_{epoch}$ | 12 | 18 | 24 | 32 | 28 | 32 | 24 |
| $b$ | 512 | 2048 | 8192 | 8192 | 8192 | 8192 | 8192 |

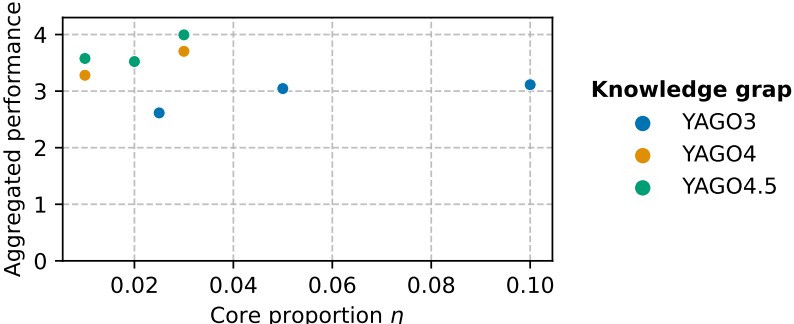

Figure 9: Effect of core proportion $\eta$ on SEPAL's performance, with the degree core selection strategy.

- **Subgraph maximum size** $m$: we use $m = 4 \cdot 10^6$. The idea is to use the largest value for which it is possible to fit the subgraph's embeddings in the GPU memory;

- **Embedding dimension** $d$: we use $d = 100$ (except for complex embeddings, where $d = 50$);

- **Number of epochs for core training** $n_{epoch}$: see Table 9;

- **Batch size for core training** $b$: see Table 9;

- **Number** $p$ **of negative samples per positive for core training**: we use $p = 100$.

### D.2 Experimental study of hyperparameter effect

Now we study experimentally the effect of varying the proportion $\eta$ of entities in the core. Figure 9 shows that increasing $\eta$ tends to improve embedding quality for downstream tasks. However, the effect seems to be plateauing relatively fast for YAGO3 (not much improvement between $\eta = 5\%$ and $\eta = 10\%$). For other datasets (YAGO4.5, YAGO4), it is not possible to explore larger values of $\eta$ because the core subgraph would not fit in the GPU memory. Moreover, decreasing $\eta$ makes SEPAL run faster, as the core embedding phase accounts for a substantial share of the total execution time (Figure 5). There is, therefore, a trade-off between time and performance.

## E  Experimental setup

**Baseline implementations.** We apply SEPAL to DistMult, and compare to PyKEEN (Ali et al., 2021c), NodePiece, PBG, and DGL-KE authors' implementations, and to FastRP. The version of NodePiece we use for datasets larger than Mini YAGO3 is the ablated version where nodes are tokenized only from their relational context. For all the baseline clustering algorithms, we used the implementations from the igraph package (Csardi, 2013) except for METIS and Spectral Clustering, for which we used the torch-sparse and scikit-learn (Pedregosa et al., 2011) implementations, respectively.

**Computer resources.** For PBG and FastRP, experiments were carried out on a machine with 48 cores and 504 GB of RAM. DistMult, DGL-KE, NodePiece, and SEPAL were trained on Nvidia V100 GPUs with 32 GB of memory. The clustering benchmark was run on a machine with 72 CPU nodes and 376 GB of RAM.

# F METHODOLOGICAL DETAILS

## F.1 METHODOLOGY FOR FIGURE 3

Figure 3 displays GPU RAM usages for SEPAL with or without BLOCS. These values are theoretical and were computed using the following procedure:

- For SEPAL with BLOCS: we took the size of the largest subgraph generated by BLOCS and computed the memory footprint of its embeddings, given their dimension ($d = 100$) and the data type used (float32).
- For SEPAL without BLOCS: we took the size of the full knowledge graph and similarly computed the memory footprint of its embeddings.

SEPAL without BLOCS on YAGO4 and YAGO4.5 could not be computed on our hardware because the embeddings of these graphs exceed our GPU memory capacity, so we had to extrapolate the values. Regarding the memory, we simply computed the requirements using the same procedure as above. Regarding the performance, we kept the same values as their with-BLOCS counterparts as the results on Mini YAGO3 and YAGO3 show that performance does not vary much when removing BLOCS.

## F.2 METHODOLOGY FOR TABLE 2

Table 2 shows the experimental compliance of several partitioning algorithms to the specific requirements of our method.

The criterion to validate requirement R1 (connected) is that all the output subgraphs have only one connected component.

For requirement R2 (bounded size), the criteria are: *1)* No subgraph should be bigger than twice the average subgraph size *2)* No subgraph should be smaller than half the average subgraph size.

Additionally, we consider both requirements to be failures for trivial partitionings: one subgraph with all the entities, or $N$ subgraphs with one entity each.

