# OpenReview forum: "SEPAL: Scalable Feature Learning on Huge Knowledge Graphs"
_ICLR.cc/2025/Conference — Submitted to ICLR 2025_

### Official Review · Reviewer_QYKs · 2024-10-30

**Soundness:** 2
**Presentation:** 3
**Contribution:** 2
**Rating:** 5
**Confidence:** 4

**Summary:**

Knowledge graphs accumulate information about relations of entities of the world. Producing high quality knowledge graph embeddings are important to many downstream applications such as knowledge graph completion, knowledge graph based entity retrieval, etc. However, existing methods hardly scale to large knowledge graphs. Works like PyTorch-BigGraph and DGL-KE leverage graph partition and distributed training to scale knowledge graph embedding (KGE) training. This paper tackles the scalability issue of KGE training from a new way. Specifically, it trains knowledge graph embeddings on a core subset of entities, which greatly reduces the computation cost and memory requirements. After training the embeddings of core entities and relations, it propagates these embeddings to the rest of the graph with message passing which does not require further embedding training. The resulting system is called SEPAL (Scalable Embedding Propagation Algorithm for Large knowledge graph).

**Strengths:**

The paper proposes a new way to train knowledge graph embedding. Instead of training embeddings on the entire KG, it only trains embeddings on a small subset of core entities of a KG, which can reduces the computation cost and memory requirements.

**Weaknesses:**

The method of constructing the core subgraph cannot guarantee that all the relation types are included in the Core subgraph. Furthermore, there is no explanation of how to select \eta to minimize the training cost while maximize the model quality (embedding quality).

The diffuse step of BLOCS can easily cause the size of a subgraph to grow significantly if the subgraph has some super nodes. How to guarantee the size of subgraph will not exceed the upper bound is not discussed.

The paper claims that the proposed method can easily be adapted to most knowledge-graph embedding method. However, in the evaluation, it only shows the training efficiency and model performance of DistMult model. The performance (system and model) of other KGE models like TransE, RotatE should be presented.

The evaluation only uses YAGO datasets. The author should also include other large scale dataset like Freebase. Both PyTorch-BigGraph and DGL-KE can train KGE on Freebase.

One important task for KGE is knowledge completion. However, the paper does not have any evaluation on KG completion tasks.

The ablation study of how different hyper-parameters, I.e., \eta, m (L259), h (L308) of SEPAL impact KGE training speed and performance is missing.

The execution time breakdown of SEPAL is missing in the evaluation. How many time it takes to produce the core subgraph? How many time it takes to training the KGE on core subgraph? How many time it takes to run BLOCS and how many time it takes to do embedding propagation.

**Questions:**

How do you define NORMALIZE in section 3.2?
How do you decide K for different datasets? Is it a hyper-parameter?
How does R2 score define?
How do you train the KGE models using PyTordh-BigGraph and DGL-KE, How do you split the training set? Have you included all the entities in Table 5 in the training set?

**Details Of Ethics Concerns:**

No.

---

> ### Author Response · Authors · 2024-11-24
> **Response to Reviewer QYKs (1/2)**
>
> We sincerely thank Reviewer QYKs for their thorough and thoughtful review, which raised several relevant and valuable points. We have carefully addressed each of these concerns individually, and provide our detailed responses below.
>
>
> - “*include other large-scale dataset like Freebase*”
>
> We agree that testing on **Freebase** is interesting, so, as suggested by the reviewer, **we added it to the paper**. The results are in the **newly added Figure 4** (page 15) and exhibit a similar pattern as other large-scale datasets, with **SEPAL outperforming prior methods**. This demonstrates the scalability of SEPAL to graphs up to 85M entities. For entity matching with the downstream tables, we used the link between YAGO and Freebase entities available in the YAGO4 files.
>
>
> - “*The performance (system and model) of other KGE models like TransE, RotatE should be presented.*”
>
> We agree with the reviewer, so **we added TransE and RotatE** to our experimental study (see Figure 4 page 15). On the “small” graph (YAGO3), where these methods can be run without SEPAL, we find that **adding SEPAL improves their performance on downstream tasks**. However, DistMult works better than TransE and RotatE for the feature enrichment task.
>
>
> - “*the paper does not have any evaluation on KG completion tasks.*”
>
> Our paper focuses on feature learning, an active research field [Grover & Leskovec 2016, Cvetkov-Iliev 2023, Robinson 2024]. While knowledge graph (KG) completion is a popular use case for embeddings, it is fundamentally different from feature enrichment. Notably, **embeddings optimized for link prediction may not perform well for feature enrichment, and vice versa**. These are different tasks; for instance, FastRP—a strong baseline for feature learning—cannot be used for KG completion as it ignores relation types. There is a blind spot in the literature as few works have studied KG embeddings for downstream tasks, despite their demonstrated utility.
>
> That being said, we did test SEPAL on link prediction, although it is not the focus of our work. It worked within the average of other methods, sometimes better, sometimes worse, depending on the dataset. However, we chose not to include these results in our paper as we did not spend time to systematically explore hyperparameter tuning for this task, which would be necessary to fairly showcase SEPAL's potential in KG completion, and compare the methods together.
>
> What our paper shows is that SEPAL consistently outperforms other methods for feature enrichment on 6 knowledge graphs. This stands independently of its performance on link prediction. We believe that having models performing well on feature enrichment is valuable for the community, and broadens the applicability of knowledge-graph embeddings.
>
> [Grover & Leskovec 2016] Grover, A., & Leskovec, J. (2016). node2vec: Scalable feature learning for networks. KDD
>
> [Cvetkov-Iliev 2023] Cvetkov-Iliev, A., Allauzen, A., & Varoquaux, G. (2023). Relational data embeddings for feature enrichment with background information. Machine Learning, 112(2)
>
> [Robinson 2024] Robinson, J., Ranjan, R., Hu, W., Huang, K., Han, J., Dobles, A., ... & Leskovec, J. (2024). Relbench: A benchmark for deep learning on relational databases. NeurIPS.
>
>
> - “*execution time breakdown of SEPAL [...] to produce the core subgraph [...] to train the KGE on core subgraph [...] to run BLOCS [...] to do embedding propagation.*”
>
> **We added Appendix A.2 and Figure 5** (page 16), which present the execution time breakdown in detail and analyze the key factors that influence it.
>
>
> - “*The diffuse step of BLOCS can easily cause the size of a subgraph to grow significantly if the subgraph has some super nodes.*”
>
> The subgraphs added by diffusion are guaranteed to be smaller than the upper bound: at line 30 of Algorithm 1 BLOCS (page 6), we append $S_{k-1}$ to the set of subgraphs, and not $S_k$. $S_k$ is the first subgraph to exceed the size limit, thus $S_{k-1}$ is smaller than the size limit. We have added a sentence in plain text to make it more apparent to the reader (lines 285 and 296).
>
>
> - “*How do you define NORMALIZE in section 3.2?*”
>
> It is a simple **L2 normalization**. In our method, this normalization is important to **accelerate the convergence** of the embeddings by canceling out the effect of neighbors still having zero-embeddings (outer embeddings are initialized with zeros).
> Additionally, normalizing embeddings is a common practice of knowledge-graph embedding models (cf. DistMult, TransE, etc.), so the outer embeddings of SEPAL remain **consistent** with those of the core.
> We added lines 343-347 in the paper to discuss this point.
>
>
> - “*The ablation study of how different hyper-parameters [...] is missing.*”
>
> We have **added Appendix D** (pages 21-22) listing all the hyperparameters and their values in our experiments. We also studied experimentally their effect on performance (**Figure 9** page 22), or explained the procedure to tune them.

---

> > ### Author Response · Authors · 2024-11-24
> > **Response to Reviewer QYKs (2/2)**
> >
> > - “*The method of constructing the core subgraph cannot guarantee that all the relation types are included in the Core subgraph.*”
> >
> > Indeed, although it is possible to increase $\eta$ until this is the case (the limit being to fit the core subgraph in GPU memory). Experimentally, the core subgraphs of our datasets contain all the relation types for reasonable choices of $\eta$, except for YAGO4. This is a convenient property of the YAGO datasets, which have very few relation types (between 37 and 78). It is probably not the case for datasets with many more relations, such as Freebase (14,665 relations).
> >
> > We designed a **new core selection technique**, based on the relations, to deal with this issue. The idea is to build the core by sampling triples with each relation type. It is presented in detail in the **new Appendix A.5** (pages 18-19).
> >
> >
> > - “*How do you decide K for different datasets? Is it a hyper-parameter?*”
> >
> > $K$, the number of message-passing steps in each subgraph, is a hyperparameter. Setting $K$ equal to the graph's diameter ensures reaching every remote entity but may slow SEPAL on graphs with long chains. In practice, choosing $K$ at 2–3 times the Mean Shortest Path Length (MSPL) usually embeds most entities effectively. We discuss this in Appendix D (pages 21-22).
> >
> >
> > - “*How does R2 score define?*”
> >
> > We have expanded Appendix C (pages 20-21), which now explains this in detail. The R2 score is the coefficient of determination, a commonly used metric for regression.
> >
> >
> > - “*How do you split the training set? Have you included all the entities in Table 5 in the training set?*”
> >
> > We use the **full knowledge graphs** to train the embeddings. Unlike link prediction, the downstream tasks involve external evaluation (predicting values not in the KG data), so we do not split the knowledge graphs into train/val/test sets.
> > Instead, the split occurs on the downstream tables before training the regression estimator. The scores we present are mean cross-validation scores, averaged over 5 repeats of 5-fold cross-validations.
> > We have **expanded Appendix C** (pages 20-21) to clarify this setup.
> >
> >
> > - “*How do you train the KGE models using PyTorch-BigGraph and DGL-KE?*”
> >
> > For PyTorch-BigGraph and DGL-KE, we use the official implementations from the authors, and the hyperparameters provided for similarly sized datasets. For both of them, we use DistMult as the base model. We included this in Appendix E (page 22) describing our experimental setup.

---

> > ### Comment · Reviewer_QYKs · 2024-11-25
> > **Response**
> >
> > Thank you to the authors for conducting additional experiments (Freebase profiling, RotatE, TransE, etc.), which have resolved some of my concerns. However, the concern regarding "The method of constructing the core subgraph cannot guarantee that all relation types are included in the core subgraph" remains unaddressed.

---

> > > ### Author Response · Authors · 2024-11-28
> > > **We added a new core selection strategy that solves this**
> > >
> > > We sincerely thank the reviewer for adjusting their evaluation.
> > >
> > > Regarding constructing the core subgraph, the reviewer is indeed right that in some cases the method in our original manuscript did not guarantee that all relation types are included (see Table 4, page 20). This prompted us to **add a new core selection method** that addresses this concern regarding core relations.
> > >
> > > We call it the “mixed” strategy because it is a mix between our two previous strategies (that in the original manuscript, and the one discussed in the previous reply), combining their two strengths: the density of degree-based core subgraphs (leading to better core embeddings), and the relational coverage of relation-based core subgraphs.
> > >
> > > Specifically, the mixed core selection strategy:
> > > - Guarantees **by design** that all the relations are in the core subgraph
> > > - Outperforms the simple relation-based strategy on downstream tasks (Figure 8, page 19), and cancels out or reduces the performance drop relative to the degree-based strategy.
> > > - Runs in a reasonable amount of time (Table 4, page 20)
> > >
> > > This new method is **described in detail in Appendix A.5.2** (pages 19-20) and **studied experimentally in the new Figure 8 and Table 4. Results show that this method indeed leaves no relation behind.** We uploaded a revised version of the paper, in which we write the changes made since our last response in green.
> > >
> > > We sincerely thank the reviewer for pointing out this problem in our original submission, as we believe the changes we have made to address it improve the robustness of our proposed method.

---

### Official Review · Reviewer_c7mu · 2024-10-31

**Soundness:** 2
**Presentation:** 2
**Contribution:** 2
**Rating:** 6
**Confidence:** 2

**Summary:**

This paper introduces SEPAL  to scale knowledge graph embedding models efficiently on limited hardware resources. SEPAL reduces computational demands by optimizing embeddings only for core subgraphs.   To facilitate efficient message passing, the authors developed BLOCS which partitions large graphs into smaller, well-connected subgraphs that fit within GPU memory. Evaluated on five knowledge graphs across four regression tasks, SEPAL demonstrates speedup over traditional embedding methods and achieves strong performance on downstream tasks.

**Strengths:**

1.This paper proposes SEPAL, a method designed to reduce computational costs, achieving faster processing times and lower memory usage for knowledge graph embeddings.

2.Although SEPAL generates embeddings from subgraphs, which could be suboptimal, the approach demonstrates strong performance on downstream tasks in experimental evaluations.

**Weaknesses:**

1.The work presents an interesting approach, though some aspects require further clarification. For instance, how specifically does the model reduce time and memory costs? The proposed method involves constructing core subgraphs and performing message passing on these subsets. In terms of memory, it appears that a substantial portion of memory usage stems from storing embeddings. However, it seems embeddings still need to be allocated for all entities and relations, raising the question of how memory savings are achieved. Regarding time reduction, is it primarily due to message passing on smaller subgraphs? Otherwise, this appears similar to conventional neighborhood-based message passing.

2.In the experimental section, the regression task setup lacks clarity. What specific values are being predicted? Additionally, why not evaluate the method on the knowledge graph completion task, which is a fundamental evaluation for knowledge graph embeddings? Including this evaluation would provide a more comprehensive assessment of the method’s performance.

**Questions:**

Please refer to the weekness

---

> ### Author Response · Authors · 2024-11-24
> **Response to Reviewer c7mu**
>
> We thank Reviewer c7mu for the careful review. We are glad that the reviewer found our proposed method interesting, despite some aspects requiring clarification. We provide more details about the specific points raised by the reviewer.
>
> **1. Time and memory reduction**
>
> **Memory reduction**: SEPAL achieves memory reductions **on the GPU**. Indeed, most of the memory usage stems from storing embeddings. While traditional KGE models load all the embeddings on the GPU, SEPAL stores them on the CPU. By loading embeddings to the GPU subgraph by subgraph, **SEPAL shifts the memory limit from GPU RAM to CPU RAM** (which is much larger), thus enabling the processing of graphs that are orders of magnitude larger. SEPAL is an **out-of-core algorithm**, to process data too large to fit into the GPU memory at once.
> We have modified Figure 3 (page 9) and lines 421 to 425 to explain more clearly how BLOCS controls the memory pressure on the GPU.
>
> **Time reduction**: SEPAL’s time reduction over traditional KGE methods comes from two key elements:
>
> - The costly triple-level embedding optimization is performed **only on the core subgraph**.
>
> - For entities outside the core, the **embeddings are optimized implicitly via message passing**, which is much quicker than proper gradient descent.
>
> We highlight that the time reduction comes from the very idea of using message passing for this purpose, and not from an improvement over conventional message passing.
>
>
> **2. Embedding evaluation**
>
> **Downstream tasks**: The experimental setup was mentioned briefly in our submission, and we thank the reviewer for pointing this out. We have expanded Appendix C (page 20-21), which describes the four downstream tasks considered in our experimental study. The high-level idea of those tasks is to take a downstream table in which entities are associated with a target value $y$, replace the entities by their embedding, and learn to predict $y$ based on the embeddings. These tasks are therefore node regression tasks.
>
>
> **No knowledge graph completion**: We focus on feature learning, an active research field [Grover & Leskovec 2016, Cvetkov-Iliev 2023, Robinson 2024]. While knowledge graph (KG) completion is a popular use case for embeddings, it is fundamentally different from feature enrichment. Notably, **embeddings optimized for link prediction may not perform well for feature enrichment, and vice versa**. These are different tasks; for instance, FastRP—a strong baseline for feature learning—cannot be used for KG completion as it ignores relation types. There is a blind spot in the literature as few works have studied KG embeddings for downstream tasks, despite their demonstrated utility.
>
> That being said, we did test SEPAL on link prediction, although it is not the focus of our work. It worked within the average of other methods, sometimes better, sometimes worse, depending on the dataset. However, we chose not to include these results in our paper as we did not spend time to systematically explore hyperparameter tuning for this task, which would be necessary to fairly showcase SEPAL's potential in KG completion, and compare the methods together.
>
> What our paper shows is that SEPAL consistently outperforms other methods for feature enrichment on 6 knowledge graphs. This stands independently of its performance on link prediction. We believe that having models performing well on feature enrichment is valuable for the community, and broadens the applicability of knowledge-graph embeddings.
>
>
> [Grover & Leskovec 2016] Grover, A., & Leskovec, J. (2016). node2vec: Scalable feature learning for networks. KDD
>
> [Cvetkov-Iliev 2023] Cvetkov-Iliev, A., Allauzen, A., & Varoquaux, G. (2023). Relational data embeddings for feature enrichment with background information. Machine Learning, 112(2)
>
> [Robinson 2024] Robinson, J., Ranjan, R., Hu, W., Huang, K., Han, J., Dobles, A., ... & Leskovec, J. (2024). Relbench: A benchmark for deep learning on relational databases. NeurIPS.

---

> > ### Comment · Reviewer_c7mu · 2024-11-27
> > **response to author**
> >
> > Thanks for the response! It address some of my concerns. But I still have the following concerns.
> >
> > 1. Regarding memory usage, if storing embeddings on the CPU can save GPU memory, would it also be feasible to use the CPU for traditional KGE methods? For example, could loading each batch into the GPU during training also achieve similar memory savings? I’d appreciate your thoughts on this.
> >
> > 2. Could you clarify how the proposed model enriches features? From my understanding, it primarily addresses the scalability issue by learning embeddings and then applies downstream tasks to train and optimize these embeddings. If the model focuses solely on learning embeddings, could you elaborate on why it might not be suitable for KGE tasks?

---

> > > ### Author Response · Authors · 2024-11-28
> > > **Response to Reviewer's concerns**
> > >
> > > We thank reviewer c7mu for the interesting questions. We address them below.
> > >
> > > **1. Memory usage**
> > >
> > > Yes, using the CPU for traditional KGE methods would be feasible, and loading each batch into the GPU during training would achieve similar GPU memory savings. However, **this would drastically increase the time cost due to data movement between CPU and GPU**. Indeed, in modern architectures, memory transfer costs are very high, amounting to much computation.
> > >
> > > Despite the increased communication cost, this approach has been explored by models such as **PBG** or **DGL-KE**, which are included in our baselines. To mitigate the time cost, they **parallelize** the training on several CPUs/GPUs/machines. This requires **preliminary graph partitioning** to prevent entities from being processed by two workers simultaneously. However, there are still **shared learnable parameters (the embeddings of the relations) that need to be synchronized regularly across all workers**, which incurs additional communication costs. Moreover, this approach introduces several functional changes that can slow down the convergence of the SGD: 1) For each minibatch, the edges are sampled within one partition 2) For each edge, the negatives are sampled within the same partition 3) Relation embeddings, which are shared between workers, are updated asynchronously.
> > >
> > > **SEPAL explores a completely different approach that does not rely on heavy parallelization, thus better suited for modest hardware.** Indeed, as discussed in our previous response, SEPAL saves time by implicitly optimizing outer embeddings through propagation, rather than explicit optimization with SGD.
> > >
> > > **2. Usage of the embeddings**
> > >
> > > - **Feature enrichment**: SEPAL solely learns knowledge-graph embeddings. For evaluation purposes, these embeddings are used for feature enrichment on downstream tasks. Feature enrichment consists of **enriching the features of a downstream table**: for a column containing known entities (e.g., cities), each entity is replaced by the embeddings learned on the knowledge graph, creating a wider table with numerical features. The embeddings are not modified at any point; they just serve as new table features to train a separate regression estimator that predicts a target value $y$ in the table. What we measure is how much a given set of embeddings helps improve the prediction of $y$, which is directly related to the amount of useful information these embeddings encapsulate, i.e., the amount of information they successfully extracted from the knowledge graph.
> > > Recent research works, such as Relbench [Robinson 2024], have clearly shown the importance of feature enrichment for learning in relational databases. It is a growing research focus, broader than KGC tasks (as it is not restricted to KGE models).
> > >
> > > - **Knowledge graph completion (KGC)**: SEPAL’s embedding could, indeed, be used for KGC tasks. However, these tasks are different from feature enrichment tasks, and they may call for **different tradeoffs**.
> > > We are currently investigating this empirically and hope to come back soon with more information (our results are currently too preliminary).
> > >
> > >
> > > We hope our responses provide clarity and address the reviewer’s concerns. Should further questions arise, we would be happy to continue the discussion.
> > >
> > > [Robinson 2024] Robinson, J., Ranjan, R., Hu, W., Huang, K., Han, J., Dobles, A., ... & Leskovec, J. (2024). Relbench: A benchmark for deep learning on relational databases. NeurIPS.

---

> > > > ### Comment · Reviewer_c7mu · 2024-11-30
> > > > **response to author**
> > > >
> > > > Thank you for the clarification. I appreciate the detailed response, but I still have the following questions:
> > > >
> > > > 1. It seems the authors employ distmult to obtain embeddings, but distmult does not require message passing. Typically, DistMult initializes node and relation embeddings and computes scores directly for optimization. Could you elaborate on how subgraph-based methods contribute and how the subgraph makes difference for distmult?
> > > >
> > > > 2. If using the CPU to store embeddings increases communication costs between the CPU and GPU, would these communication costs also affect BLOCS? How does BLOCS address or mitigate such potential bottlenecks?
> > > >
> > > > 3. It seems the METIS results are not reported in Figure 2. Since the paper addresses large graph datasets with subgraph-based methods, how does BLOCS compare to other subgraph-based approaches, such as GraphSAGE, Cluster-GCN, and GraphSAINT? What specific advantages does BLOCS offer over these methods? The novelty and contribution is not very clear compared with other subgraph-based methods.

---

> > > > > ### Author Response · Authors · 2024-11-30
> > > > > **Response to questions 1. and 2.**
> > > > >
> > > > > We thank the reviewer for their continuous effort to evaluate our work. Please find our answers to the reviewer’s questions below.
> > > > >
> > > > > **1. How SEPAL uses DistMult**
> > > > >
> > > > > - **In the core**: SEPAL **uses DistMult in the classic way**. No message passing is involved at this stage. The core subgraph typically contains 1 to 2 million entities (it is bigger than most of the knowledge graphs considered in the literature, but fits in GPU RAM), and SEPAL runs the traditional DistMult model on it. At the end of this step, we get embeddings for the core entities and for the relations, that will remain frozen until the end.
> > > > >
> > > > > - **Outside the core**: SEPAL **uses message passing to mimic DistMult, without actually training it**. The purpose is to compute outer embeddings **faster** than if we learn them with DistMult. For this, we do not run DistMult, but still leverage its fundamental relational structure: for a positive triple, the embedding of the tail should be close to the embedding of the head multiplied by the embedding of the relation, $\theta_t \simeq \theta_h \odot \theta_r$. This part is explained in section 3.2 of our paper (pages 6-7), with the equations we use for the relation-aware message passing given between L336 and L342 (page 7). During this embedding propagation, the subgraphs considered are those generated by BLOCS, and they typically contain 2 to 4 million entities (to fit in GPU RAM).
> > > > >
> > > > > We will modify the manuscript to make it more clear where the original embedding method (DistMult or another) is used.
> > > > >
> > > > > **2. Time savings of SEPAL**
> > > > >
> > > > > BLOCS itself is just the graph-division algorithm, and thus does not directly contribute to communication costs (there is no data movement at this stage). **Communication costs occur during the propagation phase**, where the embeddings of each of the subgraphs generated by BLOCS have to be loaded on the GPU, subgraph after subgraph.
> > > > > However, **these communication costs are near-optimal**.
> > > > >
> > > > > To show that, we can count the number $x$ of back-and-forth of a given embedding between CPU and GPU memory. The optimal value is $x=1$, meaning the embeddings are transferred only once. We provide a detailed analysis below:
> > > > >
> > > > > - **For core embeddings**: **core embeddings remain at all time on the GPU memory**. They are only moved to the CPU once at the end, to be saved on disk with the rest of embedings. Therefore, **the data movement of core embeddings is optimal**: $x_{core}=1$.
> > > > >
> > > > > - **For outer embeddings**: SEPAL loads each outer subgraph **only once** to the GPU. So the average number $x_{outer}$ of memory back-and-forth of outer embeddings is equal to the average number of outer subgraphs to which the corresponding outer entities belong. In the table below, we provide their experimental values for each dataset. They range from 1.2 to 7.5 depending on the dataset. **On average, each outer embedding is loaded only 2.76 times on the GPU**. To some extent, this value can be controlled by tuning the $h$ hyperparameter of BLOCS: the lower $h$, the less diffusion steps (and the more dilation steps); the less diffusion steps, the less overlap between outer subgraphs; the less overlap, the lower $x_{outer}$.
> > > > >
> > > > > |                      	| $x_{outer}$   |
> > > > > |--------------------------|-------|
> > > > > | **Mini YAGO3**       	| 1.20  |
> > > > > | **YAGO3**            	| 3.07  |
> > > > > | **YAGO4.5**          	| 7.47  |
> > > > > | **YAGO4.5 + taxonomy**   | 1.94  |
> > > > > | **YAGO4**            	| 2.29  |
> > > > > | **YAGO4 + taxonomy** 	| 1.27  |
> > > > > | **Freebase**         	| 2.09  |
> > > > > |--------------------------|-------|
> > > > > | **Average**          	| 2.76  |
> > > > >
> > > > > We stress out that the optimal value of $x=1$ can only be achieved in the case where the entire graph fits in GPU memory. Every approach that scales beyond GPU RAM limits has its communication overheads, i.e., $x>1$.
> > > > >
> > > > > For comparison, for a traditional KGE method where you would load each batch of triples on the GPU, the number of back-and-forth of an entity embedding would be: $x = n_{batches} \times n_{epochs} $, where $n_{batches}$ is the number of batches that contain a triple featuring this entity, and $n_{epochs}$ is the number of epochs.
> > > > > That is **at least one or two orders of magnitude greater than what SEPAL does in terms of data movement**. Indeed, KGE methods are usually trained for a few tens if not a few hundreds of epochs, so $x >> 10$, not even taking into account $n_{batches}$ (that can be large, depending on the graph structure).
> > > > >
> > > > > Moreover, **SEPAL also mitigates its communication overheads by using message-passing in outer subgraphs instead of a costly SGD**, as explained above.
> > > > >
> > > > > From a high-level perspective, what makes the computation fast is that:
> > > > > - it is more **memory local**: there is less back-and-forth of data across the CPU-GPU memory boundary;
> > > > > - SGD is replaced by message-passing in the outer regions of the graph.
> > > > >
> > > > > We will add a section in the final version of the paper discussing time and memory savings induced by the SEPAL framework.

---

> > > > > > ### Author Response · Authors · 2024-11-30
> > > > > > **Response to question 3.**
> > > > > >
> > > > > > **3. Comparison to other subgraph-based approaches**
> > > > > >
> > > > > > **The METIS results are reported in Figure 6** (page 17) **and discussed in Appendix A.3** (page 17) of the revised version of the paper uploaded during the discussion. We decided not to put them in Figure 2 for lack of space (and to avoid making the figure hard to read). Also, Figure 4 (page 15) summarizes all experimental results on downstream tasks, including SEPAL + METIS (i.e., replacing BLOCS with METIS in the SEPAL pipeline).
> > > > > > METIS has two main issues:
> > > > > > - It produces **disconnected** partitions, which prevents our method from propagating the core embeddings to all the outer entities;
> > > > > > - It does not scale as well as BLOCS in terms of CPU memory (Table 2, Figure 6), and thus, we could not use it for graphs larger than YAGO4.5 (32M entities).
> > > > > >
> > > > > > Regarding other subgraph-based approaches:
> > > > > >
> > > > > > **GraphSAGE** is not subgraph-based in the same sense; it samples a fixed-size neighborhood per node for inductive learning across graphs. Its adaptation to knowledge graph embedding is NodePiece, which we compare with SEPAL in Appendix A.4 (pages 17-18).
> > > > > >
> > > > > > **GraphSAINT** samples subgraphs for supervised GNN training via node classification, optimizing GNN weights without processing the full graph, while our goal is to embed the whole graph.
> > > > > >
> > > > > > **Cluster-GCN** scales GNNs by restricting the neighborhood search within precomputed clusters. It shares with SEPAL the high-level idea of breaking down the graph into smaller parts, to unlock an algorithm (GNNs for Cluster-GCN, embedding propagation for SEPAL) on larger graphs.
> > > > > >
> > > > > > While these methods train GNNs on graphs, they do not readily produce knowledge-graph embeddings on huge knowledge graphs, which is what SEPAL does. The novelty and **key contributions** of our approach lie in:
> > > > > > - The distinction between **core** and **outer** entities;
> > > > > > - The idea that **KGE models can be trained only on the core subgraph**;
> > > > > > - A **propagation algorithm**, that uses **message passing** and the **$\phi$ operator** of Table 1 (page 3), **to compute embeddings for outer entities without SGD optimization**.
> > > > > >
> > > > > > The idea of propagating a core to outer entities is not directly applicable to embed huge knowledge graphs, because their embeddings do not fit in GPU memory. Therefore we have to break down the graph and perform the propagation separately on each part. Existing partitioning algorithms are not suited for it (we benchmark them in Table 2), mainly because they do not produce connected or bounded partitions. Hence we developed our own graph-division algorithm: BLOCS, which is **another of our contributions**.
> > > > > >
> > > > > > We hope that our replies clarify our contributions and effectively demonstrate the value of our proposed method.

---

### Official Review · Reviewer_ZdA3 · 2024-11-02

**Soundness:** 2
**Presentation:** 3
**Contribution:** 2
**Rating:** 6
**Confidence:** 3

**Summary:**

This paper introduces Scalable Embedding Propagation Algorithm (SEPAL) for scalable and cost-effective training of knowledge graph (KG) embeddings over large KGs with a single GPU. Specifically, SEPAL only optimizes the embeddings for a subset of core entities, and it computes the embeddings for the rest entities with message passing. To address the scalability challenge of message passing over a large KG, it proposes an algorithm called Balanced Local Overlapping Connected Subgraphs (BLOCS) to break the KG into smaller connected subgraphs. Empirical studies demonstrate the effectiveness and efficiency of SEPAL on KGs of up to about 67M entities.

**Strengths:**

**S1.** The problem formulation and methodology design are well motivated.

**S2.** Empirical studies demonstrate the effectiveness and efficiency of SEPAL for the experiments considered.

**S3.** The paper is easy to follow.

**Weaknesses:**

**W1.** In order to defend the novelty and principled benefit of core set embedding + embedding propagation, the current discussion and comparison against previous bag-of-entities approaches, particularly NodePiece, is insufficient. Notably, NodePiece also embeds a subset of entities and the authors should discuss how this is different from SEPAL.

**W2.** Current ablation studies are insufficient. This is important in particular as both claimed contributions have counterparts explored in the existing literature, i.e., core entity embedding + embedding propagation vs bag-of-entities approaches like NodePiece and BLOCS vs graph sampling/partition like METIS. The authors should consider respectively fixing one contribution at a time and replacing the other contribution with existing approaches.

**W3.** The experiment settings considered are kind of different from the established ones without sufficient justification. See the questions below.

**Questions:**

**Q1.** Did you try evaluation for knowledge graph completion?

**Q2.** Did you try on common KGs like Freebase and Wikidata?

---

> ### Author Response · Authors · 2024-11-24
> **Response to Reviewer ZdA3 (1/2)**
>
> We sincerely thank Reviewer ZdA3 for their insightful review, helping us clarify key aspects of our work and deepen the analysis of our proposed method, ultimately improving the paper.
>
> We are delighted that the reviewer recognized our method as well-motivated, effective, and efficient for feature learning on large knowledge graphs.
>
> Below, we address the reviewer’s comments.
>
> **W1.** Indeed, SEPAL shares with bag-of-entities approaches like NodePiece that it embeds a subset of entities. Parallels can be drawn between: a) the anchors of NodePiece and the core entities of SEPAL; b) the encoder function of NodePiece and the embedding propagation of SEPAL.
> Yet, our approach **differs from NodePiece in several ways**:
>
> **Neighborhood context handling**: Both methods handle completely differently the neighborhood of entities. NodePiece tokenizes each node into a sequence of $k$ anchors and $m$ relation types, where $k$ and $m$ are fixed hyperparameters shared by all nodes.
> If the node degree is greater than $m$, **NodePiece downsamples randomly the relation tokens**, and if it is lower than $m$, [PAD] tokens are appended; both seem sub-optimal. In contrast, **SEPAL accommodates any node degree and uses all the neighborhood information**, thanks to the message-passing approach that handles the neighborhood context.
>
> Additionally, **NodePiece’s tokenization relies on an expensive BFS anchor search**, unsuitable for huge graphs. On our hardware, we could not run the vanilla NodePiece (PyKEEN implementation) on graphs bigger than Mini YAGO3 (129k entities). For YAGO3 and YAGO4.5, we had to run an ablated version where nodes are tokenized only from their relational context (i.e., $k=0$, studied in the NodePiece paper with good results), to skip the anchor search step.
>
> **Training procedure**: At train time, **NodePiece goes through the full set of triples at each epoch** to optimize both the anchors’ embeddings and the encoder function parameters, necessitating many gradient computes and resulting in long training times for large graphs.
> On the contrary, SEPAL performs mini-batch gradient descent only on the triples of the core subgraph, which provides significant time savings.
> To illustrate this, **we added Figure 7** in Appendix A.4 (page 18) which compares the performance of SEPAL and vanilla NodePiece on Mini YAGO3, showing that SEPAL outperforms NodePiece on downstream tasks while being nearly two times quicker.
>
> **Embedding propagation to non-anchor/non-core entities**: To propagate to non-anchor entities, **NodePiece has no prior knowledge of the relational structure** of the embedding space, using an encoder function (MLP or Transformer) that has to learn it through gradient descent. On the contrary, SEPAL leverages the model-specific relational structure to compute the outer embeddings with no further training needed: $\phi$ in l337 is that of the knowledge-graph embedding, as in Table 1.
>
>
> **W2.** Regarding the ablations suggested:
>
> **BLOCS vs. METIS**: Indeed, trying to replace BLOCS with METIS in our framework is an interesting experiment. **We added this in Figure 6, Appendix A.3**, page 17.
>
> However:
> 1. contrary to BLOCS, METIS outputs **disconnected** partitions (see Table 2, page 9). Given the structure of our method, this results in **zero-embeddings** for entities not belonging to the core connected component at propagation time. This theoretical argument is the reason why we did not consider implementing this experiment at first. Interestingly, the presence of zero-embeddings affects downstream scores very little, likely because most downstream entities belong to the core connected component and are thus not impacted by this.
> 2. **METIS does not scale as well as BLOCS** in terms of CPU memory. Therefore BLOCS remains an indispensable ingredient for graphs larger than YAGO4.5 (32M entities).
>
> **SEPAL without BLOCS vs. NodePiece**: New Figure 4 in Appendix A.1 (page 15) gathers all the experimental results. Among others, it enables comparing SEPAL without BLOCS to NodePiece, which reveals the superiority of our approach over NodePiece on the feature enrichment task.
> The initial version of our paper contained these results in two different figures (2 and 3). We have **modified Figure 3 to highlight the importance of BLOCS for GPU memory**.

---

> > ### Author Response · Authors · 2024-11-24
> > **Response to Reviewer ZdA3 (2/2)**
> >
> > **W3.** Regarding the experiment settings, we follow previous research on feature learning [Grover & Leskovec 2016, Cvetkov-Iliev 2023, Robinson 2024]. Considering downstream tasks allows us to compare knowledge graph embedding methods to simple but powerful baselines such as FastRP, which beats most KGE methods (cf. Figure 2 and 4) but for which KG completion is not possible (given that it does not embed relations).
> >
> > - **Q1**: “*Did you try evaluation for knowledge graph completion?*”
> >
> > Yes, we did test SEPAL on link prediction, although it is not the focus of our work. It worked within the average of other methods, sometimes better, sometimes worse, depending on the dataset. However, we chose not to include these results in our paper as we did not spend time to systematically explore hyperparameter tuning for this task, which would be necessary to fairly showcase SEPAL's potential in KG completion, and compare the methods together.
> >
> > - **Q2**: “*Did you try on common KGs like Freebase and Wikidata?*”
> >
> > As two reviewers suggested, we downloaded the Freebase knowledge graph and ran our proposed method and a few baselines on it. The results are provided in the **newly added Figure 4** (Appendix A.1, page 15) and exhibit a similar pattern as other large-scale datasets, with **SEPAL outperforming prior methods**. This **demonstrates the scalability of SEPAL to graphs up to 85M entities**. For entity matching with the downstream tables, we used the link between YAGO and freebase entities available in the YAGO4 files.
> >
> >
> > [Grover & Leskovec 2016] Grover, A., & Leskovec, J. (2016). node2vec: Scalable feature learning for networks. KDD
> >
> > [Cvetkov-Iliev 2023] Cvetkov-Iliev, A., Allauzen, A., & Varoquaux, G. (2023). Relational data embeddings for feature enrichment with background information. Machine Learning, 112(2)
> >
> > [Robinson 2024] Robinson, J., Ranjan, R., Hu, W., Huang, K., Han, J., Dobles, A., ... & Leskovec, J. (2024). Relbench: A benchmark for deep learning on relational databases. NeurIPS.

---

> > > ### Comment · Reviewer_ZdA3 · 2024-11-25
> > >
> > > Thank you for your detailed responses, which have partially addressed my concerns with solid experiment results. I've adjusted my evaluation accordingly. Meanwhile, I still feel that studying knowledge graph completion may help improve the impact of the work if possible.

---

> > > > ### Author Response · Authors · 2024-11-28
> > > > **We are working on adding KG completion**
> > > >
> > > > We sincerely appreciate that reviewer ZdA3 acknowledges the strength of our experimental results and has improved their rating accordingly.
> > > >
> > > > Regarding the suggestion to include results on knowledge graph completion, we agree with the reviewer that they would broaden the manuscript's impact. We are working on such experiments; however, our results are too preliminary to add to the paper now, and today is the last day to edit the PDF. Should the paper be accepted, we will add a complete set of experiments on KG completion in the final version.

---

> > > > > ### Author Response · Authors · 2024-12-03
> > > > > **Preliminary results on knowledge graph completion (1/2)**
> > > > >
> > > > > As suggested, we have **tested SEPAL on knowledge graph completion**.
> > > > >
> > > > > **Experimental setup**
> > > > > ----------------------------
> > > > >
> > > > > We detail our experimental setup for the link prediction task:
> > > > >
> > > > > - **Setting**: We evaluate models under the **transductive setting**: the missing links to be predicted connect entities already seen in the train graph.
> > > > >
> > > > > - **Stratification**: We randomly split each dataset into **training (90%), validation (5%), and test (5%)** subsets of triples. During stratification, we ensure that the train graph remains connected by moving as few triples as required from the validation/test sets to the training set.
> > > > >
> > > > > - **Sampling**: Given the size of our datasets, sampling is required to keep link prediction tractable. For each evaluation triple, we sample 10,000 negative entities uniformly to produce **10,000 candidate negative triples** by corrupting the positive.
> > > > >
> > > > > - **Filtering**: For tractability reasons, we report **unfiltered** results: we do not remove triples already existing in the dataset (which may score higher than the test triple) from the candidates.
> > > > >
> > > > > - **Ranking**: If several triples have the same score, we report **realistic ranks** (i.e., the expected ranking value over all permutations respecting the sort order; see PyKEEN documentation).
> > > > >
> > > > > - **Metrics**: We use three standard metrics for link prediction: the **mean reciprocal rank (MRR), hits at $k$ (for $k\in\{1,10,50\}$) and mean rank (MR)**. Given the rankings $r_1,\ldots, r_n$ of the $n$ evaluation (validation or test) triples:
> > > > > $$\mathrm{MRR} = \frac{1}{n} \sum_{i=1}^n \frac{1}{r_i},$$
> > > > >
> > > > > $$\quad\mathrm{Hits@}k = \frac{1}{n} \sum_{i=1}^n \mathbb{1}_{r_i\le k},$$
> > > > >
> > > > > $$\quad\mathrm{MR} = \frac{1}{n} \sum_{i=1}^n r_i.$$

---

> > > > > > ### Author Response · Authors · 2024-12-03
> > > > > > **Preliminary results on knowledge graph completion (2/2)**
> > > > > >
> > > > > > **Experimental results**
> > > > > > ------------------------------
> > > > > > Due to different sampling procedures, the MRR values (and other metrics) reported in the PBG and DGL-KE papers cannot be compared to ours. **To ensure comparability of results, we re-ran all the baselines on our datasets with our experimental setup**. For each one, we use DistMult as the base model.
> > > > > >
> > > > > > In the tables below, we present our results for the different metrics. They show that **SEPAL is competitive with existing methods** (DGL-KE, PBG) and that no method is consistently better than the others for all datasets.
> > > > > >
> > > > > > These are **preliminary** results; experiments on large graphs and evaluation take time. There is still hyperparameter setting to work on, leaving room for improvements in SEPAL’s performance.
> > > > > > We started with the hyperparameters provided in Table 9 (page 23).
> > > > > >
> > > > > > Contrary to downstream tasks, the mixed core selection strategy yields better results than its degree-based counterpart. This highlights **different trade-offs between downstream tasks and link prediction**. For link prediction, good relational coverage seems to count most, whereas for downstream tasks, the core density matters most (Figure 8, Appendix A.5, pages 18-19).
> > > > > >
> > > > > > - **Mean reciprocal rank (MRR)**: higher is better
> > > > > >
> > > > > > |              	| YAGO3   | YAGO4.5 | YAGO4.5 + taxonomy | YAGO4   | YAGO4 + taxonomy | Freebase | Average |
> > > > > > |------------------|---------|---------|--------------------|---------|------------------|----------|---------|
> > > > > > | DistMult     	| 0.8049  | -   	| -              	| -   	| -            	| -    	| -   	|
> > > > > > | NodePiece    	| 0.2596  | 0.4456  | -              	| -   	| -            	| -    	| -   	|
> > > > > > | PyTorch-BigGraph | 0.5581  | 0.5539  | 0.5688         	| 0.6406  | 0.6224       	| 0.7389   | 0.6138  |
> > > > > > | DGL-KE       	| 0.7284  | 0.62	| 0.6469         	| 0.2372  | 0.257        	| 0.3017   | 0.4652  |
> > > > > > | SEPAL        	| 0.5752  | 0.5417  | 0.5357         	| 0.4726  | 0.477        	| 0.5378   | 0.5233  |
> > > > > >
> > > > > > - **Hits@1**: higher is better
> > > > > >
> > > > > > |              	| YAGO3   | YAGO4.5 | YAGO4.5 + taxonomy | YAGO4   | YAGO4 + taxonomy | Freebase | Average |
> > > > > > |------------------|---------|---------|--------------------|---------|------------------|----------|---------|
> > > > > > | DistMult     	| 0.74	| -   	| -              	| -   	| -            	| -    	| -   	|
> > > > > > | NodePiece    	| 0.1735  | 0.3449  | -              	| -   	| -            	| -    	| -   	|
> > > > > > | PyTorch-BigGraph | 0.5 	| 0.4939  | 0.4977         	| 0.5494  | 0.5416       	| 0.7015   | 0.5473  |
> > > > > > | DGL-KE       	| 0.6663  | 0.5511  | 0.5733         	| 0.1642  | 0.1892       	| 0.2498   | 0.399   |
> > > > > > | SEPAL        	| 0.4603  | 0.4819  | 0.4877         	| 0.3746  | 0.3755       	| 0.4824   | 0.4437  |
> > > > > >
> > > > > > - **Hits@10**: higher is better
> > > > > >
> > > > > > |              	| YAGO3   | YAGO4.5 | YAGO4.5 + taxonomy | YAGO4   | YAGO4 + taxonomy | Freebase | Average |
> > > > > > |------------------|---------|---------|--------------------|---------|------------------|----------|---------|
> > > > > > | DistMult     	| 0.9059  | -   	| -              	| -   	| -            	| -    	| -   	|
> > > > > > | NodePiece    	| 0.4388  | 0.6379  | -              	| -   	| -            	| -    	| -   	|
> > > > > > | PyTorch-BigGraph | 0.6562  | 0.6642  | 0.7021         	| 0.803   | 0.7662       	| 0.8053   | 0.7328  |
> > > > > > | DGL-KE       	| 0.8293  | 0.7446  | 0.7821         	| 0.3786  | 0.3842       	| 0.3964   | 0.5859  |
> > > > > > | SEPAL        	| 0.775   | 0.6467  | 0.6183         	| 0.6573  | 0.6778       	| 0.6398   | 0.6691  |
> > > > > >
> > > > > > - **Hits@50**: higher is better
> > > > > >
> > > > > > |              	| YAGO3   | YAGO4.5 | YAGO4.5 + taxonomy | YAGO4   | YAGO4 + taxonomy | Freebase | Average |
> > > > > > |------------------|---------|---------|--------------------|---------|------------------|----------|---------|
> > > > > > | DistMult     	| 0.9504  | -   	| -              	| -   	| -            	| -    	| -   	|
> > > > > > | NodePiece    	| 0.7358  | 0.8112  | -              	| -   	| -            	| -    	| -   	|
> > > > > > | PyTorch-BigGraph | 0.7259  | 0.7734  | 0.8136         	| 0.8891  | 0.8514       	| 0.8541   | 0.8179  |
> > > > > > | DGL-KE       	| 0.8873  | 0.8171  | 0.8748         	| 0.6037  | 0.5968       	| 0.5547   | 0.7224  |
> > > > > > | SEPAL        	| 0.886   | 0.7546  | 0.7107         	| 0.7661  | 0.8068       	| 0.7476   | 0.7786  |
> > > > > >
> > > > > > - **Mean rank (MR)** over 10,000 candidates: lower is better
> > > > > >
> > > > > > |              	| YAGO3  | YAGO4.5 | YAGO4.5 + taxonomy | YAGO4  | YAGO4 + taxonomy | Freebase | Average |
> > > > > > |------------------|--------|---------|--------------------|--------|------------------|----------|---------|
> > > > > > | DistMult     	| 64.19  | -   	| -              	| -  	| -            	| -    	| -   	|
> > > > > > | NodePiece    	| 154.7  | 263.2   | -              	| -  	| -            	| -    	| -   	|
> > > > > > | PyTorch-BigGraph | 820.9  | 408 	| 300.3          	| 117.1  | 203.5        	| 243.4	| 348.9   |
> > > > > > | DGL-KE       	| 187.7  | 624.1   | 219.9          	| 224.9  | 271.5        	| 227  	| 292.5   |
> > > > > > | SEPAL        	| 131.3  | 290.7   | 373.3          	| 553	| 363.2        	| 357.3	| 344.8   |

---

### Author Response · Authors · 2024-12-03
**Global summary of the discussions and new elements**

We thank all reviewers for their careful reading of our work and their thoughtful feedback. All reviewers acknowledged that our work reduces the computation cost and memory requirements of training embedding models on huge knowledge graphs while demonstrating strong performance on downstream tasks.

Our paper provides a novel perspective for knowledge-graph embedding: optimizing the embeddings only on a small core of entities, and then using message passing to mimic the underlying embedding model and efficiently propagate to the rest of the graph. It also contributes a technical solution, BLOCS, that allows this approach to be scaled to huge graphs within the GPU RAM limits.

---------------

Reviewers posed insightful questions and constructive comments, which we have addressed in detail in our individual responses. These have helped us improve the quality of our work, and we sincerely thank the reviewers for that.

We provided a new version of the paper with the following main additions (in blue in the paper):
- New experiments on **Freebase**, a dataset with 85M entities and 338M triples, show that **on huge graphs SEPAL creates the best embeddings for downstream tasks and is fastest** (Figure 4, page 15).
- We showed empirically that SEPAL adapts well to **other embedding models (TransE and RotatE), making them faster and better performing** (Figure 4, page 15).
- We implemented a new core selection method that **guarantees by design that all relation types are included in the core**, addressing reviewer QYKs’ concern (Appendix A.5, pages 18-19).
- We added experimental results:
  - breakdown of execution time (Figure 5, Appendix A.2, page 16)
  - effect of hyperparameters (Figures 8 and 9, Appendix D, pages 22-23)
  - ablation study replacing BLOCS with METIS (Figure 6, Appendix A.3, page 17)
  - and comparison to NodePiece (Figure 7, Appendix A.4, pages 17-18).
- We expanded Appendix C on downstream tasks (pages 21-22), clarifying our experimental setup.
- And many minor additions or clarifications suggested by the reviewers.

The discussion with reviewer c7mu led us to analyze in detail the **origin of time and memory gains** of SEPAL, and we will include this analysis in the final version of the paper.

----------------

Finally, we wanted to discuss here the question of **knowledge graph completion**, as it was raised by the three reviewers.

Our paper focuses on feature learning, an active research field [Grover & Leskovec 2016, Cvetkov-Iliev 2023, Robinson 2024]. While knowledge graph (KG) completion is a popular use case for embeddings, it is fundamentally different from feature enrichment. Notably, embeddings optimized for link prediction may not perform well for feature enrichment, and vice versa. These are different tasks; for instance, FastRP—a strong baseline for feature learning—cannot be used for KG completion as it ignores relation types. There is a blind spot in the literature, as few works have studied KG embeddings for downstream tasks, despite their demonstrated utility.

Our paper shows that SEPAL consistently outperforms other methods for feature enrichment on 6 knowledge graphs. This stands independently of its performance on link prediction. We believe that having models performing well on feature enrichment is valuable for the community, and broadens the applicability of knowledge-graph embeddings.

Nevertheless, **we conducted additional experiments to test SEPAL on knowledge graph completion**. These **preliminary results show that SEPAL can also perform well for link prediction**. We detail them in the response to reviewer ZdA3.


[Grover & Leskovec 2016] Grover, A., & Leskovec, J. (2016). node2vec: Scalable feature learning for networks. KDD

[Cvetkov-Iliev 2023] Cvetkov-Iliev, A., Allauzen, A., & Varoquaux, G. (2023). Relational data embeddings for feature enrichment with background information. Machine Learning, 112(2)

[Robinson 2024] Robinson, J., Ranjan, R., Hu, W., Huang, K., Han, J., Dobles, A., ... & Leskovec, J. (2024). Relbench: A benchmark for deep learning on relational databases. NeurIPS.

---

### Meta-Review · Area_Chair_VEPJ · 2024-12-21

**Metareview:**

This paper proposes a Scalable Embedding Propagation Algorithm (SEPAL) for scalable and cost-effective training of knowledge graph (KG) embeddings over large KGs with a single GPU. The key idea is to only optimize embeddings on a core subset of entities and compute the embeddings for the rest of the entities with message passing. To address the scalability challenge of message passing over a large KG, an algorithm called Balanced Local Overlapping Connected Subgraphs (BLOCS) is proposed to break the KG into smaller connected subgraphs.

The initial submission required extensive content additions to address the reviewers' concerns, and the revised version of the paper is substantially different from the initial version; thus, another round of reviews is needed to decide whether the revised paper is ready for publication. For example, Reviewer QYKs expressed remaining concerns regarding "The method of constructing the core subgraph cannot guarantee that all relation types are included in the core subgraph", and the authors added "a new core selection strategy", which should be one of the main components of the algorithm. Such a revision changes the core algorithm and, thus, should be described in the main section instead of the appendix. Also, the additional experiments added during the rebuttal and discussion phases had to be included in the initial submission. While the authors' efforts in improving the paper are appreciated, the initial submission is incomplete, and the revised algorithm and added experiments should be re-evaluated.

**Additional Comments On Reviewer Discussion:**

Reviewer c7mu indicated low confidence, whereas Reviewer QYKs indicated high confidence with a detailed review. Reviewer QYKs pointed out a critical issue regarding including all relation types in the core subgraph, which caused the authors to introduce a new strategy to resolve this issue. Overall, the authors had to add substantially more content to the initial submission to address the reviewers' concerns, resulting in a pretty different version from the initial submission.

---

### Decision · Program_Chairs · 2025-01-22

Reject